# Electrocardiogram Fiducial Point Detector Using a Bilateral Filter and Symmetrical Point-Filter Structure

**DOI:** 10.3390/ijerph182010792

**Published:** 2021-10-14

**Authors:** Tae-Wuk Bae, Kee-Koo Kwon, Kyu-Hyung Kim

**Affiliations:** Daegu-Gyeongbuk Research Center, Electronics and Telecommunications Research Institute, Daegu 42994, Korea; kwonkk@etri.re.kr (K.-K.K.); jaykim@etri.re.kr (K.-H.K.)

**Keywords:** bilateral filter, ECG, fiducial points, MIT-BIH database, PQRST

## Abstract

The characteristics or aspects of important fiducial points (FPs) in the electrocardiogram (ECG) signal are complicated because of various factors, such as non-stationary effects and low signal-to-noise ratio. Due to the various noises caused by the ECG signal measurement environment and by typical ECG signal deformation due to heart diseases, detecting such FPs becomes a challenging task. In this study, we introduce a novel PQRST complex detector using a one-dimensional bilateral filter (1DBF) and the temporal characteristics of FPs. The 1DBF with noise suppression and edge preservation preserves the P- or T-wave whereas it suppresses the QRS-interval. The 1DBF acts as a background predictor for predicting the background corresponding to the P- and T-waves and the remaining flat interval excluding the QRS-interval. The R-peak and QRS-interval are founded by the difference of the original ECG signal and the predicted background signal. Then, the Q- and S-points and the FPs related to the P- and T-wave are sequentially detected using the determined searching range and detection order based on the detected R-peak. The detection performance of the proposed method is analyzed through the MIT-BIH database (MIT-DB) and the QT database (QT-DB).

## 1. Introduction

Arrhythmias occur when electrical signals to the heart that coordinate heartbeats disfunction, and most arrhythmias are harmless. However, in case they are particularly abnormal or caused by a harmed heart, they may cause serious and even potentially fatal symptoms [1]. Arrhythmia is also called cardiac dysrhythmia or irregular heartbeat. Most arrhythmias are not severe, but some may predispose the individual to stroke or cardiac arrest. Globally, cardiovascular diseases are the biggest death-related factor [2].

The electrocardiogram (ECG) has become a very important bio-signal for the prompt detection of cardiovascular disease including arrhythmia. In the ECG signal, QRS complex detection is a preceding and essential step for classifying successive heartbeat rhythm. However, the measured ECG waveforms are deformed because of baseline wander, electrode motion artifact, and muscle artifact, thus making accurately detecting the QRS-interval a challenging task [3,4].

Many efforts have been given to the detection of the R-peak or QRS-interval under various noises. A well-known QRS detection method is the Pan–Tompkins algorithm [5]. Furthermore, QRS detection methods can be classified into differentiation [6], designed filters [5,7,8,9], Wavelet transform [10,11], neural networks [12], Hilbert transform [13,14], phasor transform [15], adaptive threshold [16,17], morphology [18,19], signal energy [5,20,21,22,23,24,25], and high-order moments [26] based algorithms. The difference-based method was used to identify fiducial points (FPs) in [27]. In [28], a dynamic threshold with a finite state machine was used for R-peak detection. In [29], the differentiation techniques were used to detect FPs. Two-stage median filter-based approaches [30,31] were used for baseline wandering elimination and signal deformation caused by motion artifacts.

Several PQRST complex detection methods have been proposed to date; however, they have not been extensively studied or compared to QRS detection techniques. The first reason is that P- or T-wave detection is not easy due to the characteristics of the P- or T-wave signal; the signals have a low slope, low magnitude, and various widths. This means that reliable P- or T-wave detection should be preceded by the detection of QRS or the R-point, which has a pronounced signal characteristic compared to the P- or T-wave. This means that a high-performance R-peak or QRS detection technique must be preceded for reliable P- or T-wave detection. Furthermore, abnormalities in the R-, P-, and T-waves make detection of relevant FPs difficult. The second reason is the lack of a universal definition of the FPs of each wave. Thus, a new definition of FPs for each algorithm is required for detecting ECG FPs. This also leads to a mapping problem of undetected points when no onset or offset point is detected in a particular wave of one ECG signal. Furthermore, this problem affects the heart rate variability (HRV), which is used to analyze the patient’s condition using almost all FPs. Therefore, the detection definition for FPs is not simple. Nevertheless, many different PQRST complex detection approaches based on (1) filtering techniques, such as adaptive filtering [32], low-pass differentiation (LPD) [33], nested median filtering [34], extended Kalman filtering [35,36], and switching Kalman filter [37]; (2) transform coefficient techniques, such as discrete Fourier transform (DFT) [38], discrete cosine transform (DCT) [39], wavelet [40,41,42], and hybrid feature extraction algorithm (HFEA) [42]; and (3) inference model techniques, such as Bayesian [43,44,45], hidden Markov model [46,47], and partially collapsed Gibbs Sampler (PCGS) [44,45], can be found in the literature. These methods exhibit excellent performance in identifying ECG FPs. Recently, a 12 lead-based local distance transformation (TLBLDT) was proposed for ECG wave detection and delineation [48]. However, the studies did not perform HRV analysis based on the detected ECG FPs for some medical diagnosis or pathology analysis. In addition, most methods have a high computational complexity and do not describe real-time processing techniques for wireless mobile healthcare applications, which is a recent issue. In addition, the lack of abundant detection results for FPs on the databases used in each method impairs the understanding of the algorithm.

The propagation of an action potential through the heart creates an electric current, which can be sensed at the body surface. ECG records these changes and the PQRST complex to designate the waves in the ECG, which provides valuable information in cardiac disease diagnosis. In this study, we introduce a novel PQRST complex detector that detects all the FPs on the ECG signal by detecting the QRS complex using the characteristics of a one-dimensional bilateral filter (1DBF), reducing the noise and enhancing the sharpness, and then, recognizing the P- and T-wave based on the detected QRS. This method first detects the R-peak and QRS complex using the difference between original and filtered ECG by the 1DBF, and then, recognizes the onset and offset points of the remaining P- and T-waves according to a predetermined order while moving in the left–right direction based on the detected R-peak and QRS complex. The resulting location accuracy of the R-peaks detected by the proposed method are then analyzed and time- and frequency-domain analysis is performed. The remaining FPs are used to estimate trends in the ST-segment, PR-interval, and QT-interval to estimate cardiac status.

## 2. Materials and Methods

### 2.1. DBF and Its Application

The 1DBF proposed is a nonlinear filter that smooths noises mixed into a signal while preserving edge features of a signal [49,50]. The filtering operation of the 1DBF restores a degraded signal. The normalization factor of the 1DBF guarantees that the filter conserves the average of flat areas in a signal.

The 1DBF is composed of two low-pass Gaussian filters (LPGFs) for the domain and range space. The domain low-pass Gaussian filter (DLPGF) assigns more weight to signal values spatially closer to the centered value. The range low-pass Gaussian filter (RLPGF) provides more weight to signal values closer to the value of the centered value. The 1DBF at an edge region, such as QRS complex, becomes a lengthened Gaussian filter. This ensures that averaging is mostly operated along the edge and is significantly decreased in the direction of the gradient. Therefore, the 1DBF can smooth noises while conserving the edge region. To use 1DBF as an efficient ECG background predictor except the QRS complex, the standard deviations,
$\sigma_{d}$
and
$\sigma_{r}$, of the domain and range Gaussian filters must be adaptively changed.

Figure 1 shows the impact of standard deviations of two types of Gaussian filters using 215N(4) in MIT-DB. Here, 215N(4) represents that the fourth annotated heartbeat belongs to N (Normal) class for the Record 215.
$\sigma_{d}$
controls the width of the domain filter. A smaller
$\sigma_{d}$
leads to weak smoothing, and a smaller
$\sigma_{r}$
works so that the range filter dominates in the two Gaussian filters in the 1DBF. Figure 1b shows that 1DBF with a little bigger
$\sigma_{d}$
removes noises and preserves the edge structures for the degraded ECG signal shown in Figure 1a. In Figure 1c, where the domain Gaussian filter is dominantly applied with a bigger
$\sigma_{r}$, the QRS complex corresponding to the edges is significantly blurred. In Figure 1d, a bigger
$\sigma_{d}$
makes the QRS complex more blurred under the same
$\sigma_{r}$. Based on this experiment, for 1DBF to act as a background predictor, which preserves the non-QRS-interval while suppressing the QRS-interval, bigger
$\sigma_{d}$
and $\sigma_{r}$
should be applied for the QRS-interval, whereas bigger
$\sigma_{d}$
and $\sigma_{r}$
should be applied for the non-QRS-interval.

### 2.2. Proposed ECG PQRST Complex Detector Using 1DBF and Temporal Characteristics of FPs

#### 2.2.1. Detection Order of PQRST FPs

Figure 2 shows the FPs and their detection order related to the PQRST complex in the study. A general ECG waveform is composed of a P-wave (0.08–0.10 s), QRS-wave (0.06–0.10 s), and T-wave (0.16 s). The QRS complex is narrower and taller compared to the P- and T-waves. The P-, Q-, R-, S-, and T-point are denoted by the red dotted circles, and the blue dotted circles represent the onset and offset points. The detection order of the FPs was decided to consider the signal characteristics of the points, i.e., ease of detection. The ST-segment, and PR- and QT-interval analyzed in this study are also shown in Figure 2.

#### 2.2.2. Algorithm Structure for Real Time

Figure 3 shows a seamless connection structure of successive sliding windows (SWs) for real time. The first SW includes six beats, and the first and sixth beat of these are incomplete beats for which some FPs are retrieved, whereas the second through the fifth beat are perfect beats for which all PQRST points can be detected. The R-peak locations of the first and sixth beat in the first SW are set to the lower and upper bounds for the FP detection of the five perfect beats. In the second SW, the lower and upper bounds correspond to the fourth and ninth beat. Furthermore, the fifth to eighth beats are perfect beats. Therefore, the fifth ECG beat is the common ECG beat in the first and the second SW. That is, the detected PQRST points in the fifth beat are a reference connecting the points detected in the preceding and following SWs. The number of seam beats may be plural for a high heart rate (HR). In the proposed method, the SW length is 2000 samples based on MIT-DB (360 Hz), and the step size, the distance between windows, is 1600 samples.

#### 2.2.3. R-Peak Detection Using 1DBF

Original ECG signals suffer from noise and dc wandering. To detect R-peaks or QRS complexes while avoiding this problem, we use the difference of the original signal and filtered signal by 1DBF. FP detection of the proposed method is processed in units of SWs. First, the original ECG sequence S for the SW is normalized as follows:(1)$$I=(S-S_{\min})/S_{\max}$$
where I is the normalized signal for the original sequence and $S_{\min}$ and $S_{\max}$ are the min and max value in the original sequence, respectively. The length of I and S is equal to the SW.

In the original ECG signal, the QRS complex is narrow and high, whereas the P- and T-wave except for the QRS complex and the baseline are relatively wide and gentle. A background ECG signal excluding the QRS complex is predicted by a 1DBF, and a QRS complex is detected using the difference between the original ECG signal (I) and the predicted background ECG signal (B) as the following:(2)QRS=I−B

In this study, we employed a 1DBF to predict the background ECG signal (excluding the QRS signal) for the normalized original ECG signal. The background ECG signal B for the normalized original ECG signal I is given as follows:(3)$$B(x)=\frac{1}{W_{p}}\sum_{x_{i}\in \Omega }I(x_{i})\exp\left(-\frac{{\left(x_{i}-x\right)}^{2}}{2\sigma_{d.\Omega }^{2}}\right)\exp\left(-\frac{{\left(I(x_{i})-I(x)\right)}^{2}}{2\sigma_{r.\Omega }^{2}}\right)$$
where $\Omega =\{x_{i}|x-N\leq x_{i}\leq x+N\}$. Ω represents the size of the filter window based on the ECG sample-point coordinate x currently processed. The filter size is 1 × 15, which is set based on the average QRS complex time (0.06–0.10 s) and the sampling rate (360 Hz) of MIT-DB. The standard deviations, $\sigma_{d.\Omega }$ and $\sigma_{r.\Omega }$, of DLPGF and RLPGF applied to the 1DBF using the variance $\sigma^{2}(\Omega )$ of the sample points in the filter window are given as follows:(4)$$\sigma_{d.\Omega }=\log_{2}\left({\left(\frac{\sigma^{2}(\Omega )}{{\left(\sigma^{2}\right)}_{\max}}+1\right)}^{{\left(\sigma_{d}\right)}_{\max}}\right)+\sigma_{\min}$$
(5)$$\sigma_{r.\Omega }=\log_{2}\left({\left(\frac{\sigma^{2}(\Omega )}{{\left(\sigma^{2}\right)}_{\max}}+1\right)}^{{\left(\sigma_{r}\right)}_{\max}}\right)+\sigma_{\min}$$
where the maximum variance is ${\left(\sigma^{2}\right)}_{\max}=0.1310$ and was obtained experimentally for the MIT-DB. The maximum standard deviations of DLPGF and RLPGF are ${\left(\sigma_{d}\right)}_{\max}=10$ and ${\left(\sigma_{r}\right)}_{\max}=1$, respectively. Here, $\sigma_{\min}=10^{-5}$ is the minimum standard deviation to prevent the two standard deviations from becoming zero when $\sigma^{2}(\Omega )=0$. The above two equations for obtaining the background ECG signal for the original ECG signal mean the following: QRS regions with high variance are smoothed by bigger $\sigma_{d}$ and $\sigma_{r}$, whereas non-QRS regions with low variance are conserved by smaller $\sigma_{d}$ and $\sigma_{r}$.

The normalization factor is given by:(6)$$w_{p}=\sum_{x_{i}\in \Omega }\exp\left(-\frac{{\left(x_{i}-x\right)}^{2}}{2\sigma_{d}^{2}}\right)\exp\left(-\frac{{\left(I(x_{i})-I(x)\right)}^{2}}{2\sigma_{r}^{2}}\right)$$

Then, the residual (difference) signal D between the original and predicted background ECG signal is given by:(7)$$D(x)=\left|I(x)-B(x)\right|$$

The enhanced residual signal E, which improves the residual signal, is given by:(8)$$E(x)=\sum_{i=x-n}^{x+n}D(i)$$

Thereafter, the candidate-R interval $C_{R}(x)$, which has a value of 1 for the candidate-R interval and 0 for the remaining interval, is calculated as follows using the threshold:(9)$$C_{R}(x)=\left\{\begin{array}{l} 1,\;\;\;\mathrm{if}\;E(x)>{\mathrm{Th}}_{c}\\ 0,\;\;\;\mathrm{otherwise} \end{array}\right.$$
where the threshold for detecting the candidate-R interval is ${\mathrm{Th}}_{c}=E_{\max}/2$, and $E_{\max}$ is the maximum value in the difference signal of the SW size. It is most probable that the R-peak is likely to be contained in rows of ones in $C_{R}$ as shown in Figure 4. If the number of consecutive 1s is more than 8 (L≥8), it is assumed that an R-peak exists in it. An $\hat{x}$ value with a max or min value in the original signal corresponding to the rows of ones in $C_{R}$ denotes the location of the R-peak as the following:(10)$$R(=\hat{x})=\left\{\begin{array}{l} \underset{i\;\in \;\{i|C_{R}(i)=1,\;L\geq 8\}}{\arg\max}\{I(i)\}(\gamma =+1),\;\;\mathrm{if}\;(\mathrm{Max}(I(i))-I_{\mathrm{mean}})\geq (I_{\mathrm{mean}}-\mathrm{Min}(I(i)))\\ \underset{i\;\in \;\{i|C_{R}(i)=1,\;L\geq 8\}}{\arg\min}\{I(i)\}(\gamma =-1),\;\;\mathrm{if}\;(I_{\mathrm{mean}}-\mathrm{Min}(I(i)))>(\mathrm{Max}(I(i))-I_{\mathrm{mean}}) \end{array}\right.$$
where Max(I(i)) and Min(I(i)) are the max and min values, respectively, in the rows of ones, L≥8. $I_{\mathrm{mean}}$ is the mean of the original ECG signal for the currently processed SW. In Equation (10), the above conditional statement is used to detect a positive R-wave, which is a normal R-wave, and the following conditional statement is used to detect a negative R-wave, such as premature ventricular contraction (PVC). Furthermore, γ is an R-peak type factor representing a positive R-peak (γ=+1) or negative R-peak (γ=−1).

Figure 5 shows the R-peak detection procedure using the background signal estimation method described above. In Figure 5a, the blue and red lines represent the original and predicted background signal, respectively. In the predicted background signal, the QRS-interval regarded as noise is suppressed; however, the remaining signal is restored close to the original signal. Figure 5b shows the residual (difference) signal between the original and predicted background signal, and Figure 5c shows the enhanced residual signal. The candidate-R interval is detected using the threshold for the enhanced residual signal, and the maximum value in the original signal corresponding to the candidate-R interval corresponds to the R-peak, as shown in Figure 5d. If there are multiple equal maxima in the QRS-interval (i.e., when the R-peak is flat, not a point), the position of the R-peak is set to their average position.

#### 2.2.4. Detection of an Abnormal R-Peak

The shapes of the R-peak considered in this study are shown in Figure 6. The normal R-peak shown in Figure 6a is very easy to process using the algorithm because there is only one maximum value corresponding to an R-peak in the candidate-R interval. However, in addition to the normal R shape, various R-waves may exist depending on the cardiac state of the patient. In the flat R-wave shown in Figure 6b, the average position of the maximum values in the candidate-R interval is assigned to the R-peak location. Figure 6c through 6f show various notch (fragmented) R-waves. The presence of a fragmented R-wave is very important because it is related to an increase in adverse cardiac events. R-peak detection in the fragmented R-wave, such as fR-L, fR-R, fR-L2, and fR-R2, involves detection of the simple maximum value (red circles in Figure 6) in the candidate-R interval; however, the determination of the fragmented R-wave and additional R-peaks (blue circles in the figure) require further processing in the Q- or S-wave detections.

The detection of additional R-peaks is important not only to determine the fragmented R-wave but also to detect accurate Q- and S-wave FPs. Referring to the shapes of the fragmented R-peaks in Figure 6, additional R-peaks are detected for the left and right specific intervals based on the detected R-peak. First, an additional R-peak $R_{R}$ on the right side of the detected R-peak is detected by the following slope change and threshold condition for the 22-point section from the adjacent right location R+1 of the detected R-peak:(11)$$\begin{array}{l} I(x)=R_{R}\;\mathrm{if}\;\left\{\begin{array}{l} \gamma \times \mathrm{sign}\left(I^{\prime }((x-1)+n)\right)\leq 0\;\;\mathrm{and}\\ \gamma \times \mathrm{sign}\left(I^{\prime }(x+n)\right)\geq 0\;\;\mathrm{and}\\ \sum_{m=1}^{10}\left|I(x+n+(m-1))-I((x+1)+n+m)\right|>{\mathrm{Th}}_{f} \end{array}\right.\;\\ \;\;\;\;\;\;\;\;\;\;\mathrm{for}\;R(\mathrm{or}\;R_{R})+1\leq n\leq R(\mathrm{or}\;R_{R})+\alpha_{f} \end{array}$$
where $I^{\prime }(t)$ denotes the difference of I(t)−I(t+1) at any time t and $\alpha_{f}(=(1/2)\times F_{s}\times \mathrm{QRS})$ is the SR of the additional R-peak. QRS(=0.12 sec) is the typical QRS time and $F_{s}(=360\;\mathrm{Hz})$ is the sampling rate. Because the difference between the sampling points is large around the fragmented R-wave, if the sum of the differences between the 10 points to the right of the current point is more than or equal to the threshold (${\mathrm{Th}}_{f}=0.1$), that point is considered an additional R-peak. The threshold is applied for normalized signals. By recursively searching for additional R-peaks that satisfy the above conditions within the SR, abnormal R-waves with multiple additional R-peaks can be identified, as shown in Figure 6e,f.

Similar to the additional R-peak on the right, an additional R-peak $R_{L}$ on the left side can be detected by the following slope change and the threshold condition for the 22-point section from the adjacent left location R−1 of the detected R-point:(12)$$\begin{array}{l} I(x)=R_{L}\;\mathrm{if}\;\left\{\begin{array}{l} \gamma \times \mathrm{sign}\left(I^{\prime }((x-1)+n)\right)\leq 0\;\;\mathrm{and}\\ \gamma \times \mathrm{sign}\left(I^{\prime }(x+n)\right)\geq 0\;\;\mathrm{and}\\ \sum_{m=1}^{10}\left|I((x-1)+n-m)-I(x+n+(m-1))\right|>{\mathrm{Th}}_{f} \end{array}\right.\\ \;\;\;\;\;\;\;\;\;\;\mathrm{for}\;R(\mathrm{or}\;R_{L})-\alpha_{f}\leq n\leq R(\mathrm{or}\;R_{L})-1 \end{array}$$

The MIT-DB includes 48 half-hour recordings with 360 Hz, and the heartbeat is classified and labeled into 15 types. The types of heartbeats belong to five classes as shown in Table 1 [50,51]. The abbreviated representation of the ECG signal interval is used for the experiment. 111L(320) denotes that the 320-th annotated beat of the Record 111 corresponds to L (Left Bundle Branch Block, LBBB) among class N mentioned in Table 1.

Fragmented R-waves can have one or more additional R-peak, depending on the patient’s condition. The additional R-point is detected as the point farthest from the detected R-peak, which satisfies the condition of Equation (9) or Equation (10) for SR. Figure 7 shows the detection results of R-peaks and additional R-peaks in the MIT-DB, as mentioned in Figure 6. In Figure 7, fR-L and fR_R denote the fragmented R-peak to the left and right of the detected R-peak, respectively.

#### 2.2.5. SR of FPs Related to P- and T-Waves

The reliable detection SR of FPs included in the P- and T-wave is introduced in Figure 8. In the SR of all FPs of one ECG beat including the currently detected R-peaks, the lower and upper limit are the locations of previous T-offset and the next R-peak. The SR for P-point detection corresponds to the previous T-offset through the current Q-onset. Likewise, the SR for T-point detection corresponds to the current S-offset through the next R-peak position.

The proposed method considers the mapping problem of the undetected points when not even some or all FPs of the P- or T-wave are discovered. In case the P-point (T-point) is not discovered as a prerequisite, neither P-onset (T-onset) nor P-offset (T-offset) can be assigned. In this case, all FPs concerning the P-wave (T-wave) are mapped to a point Q-onset (S-offset). In case only P-point (T-point) is detected, non-detected P-onset (T-onset) is mapped to the P-point (S-offset). Likewise, in case P-offset (T-offset) is not discovered under the detected P-point (T-point), the non-detected P-offset (T-offset) can be mapped to Q-onset (T-point). This point-mapping method reduces the large error caused by undetected FPs at the calculation of the PR-interval or PR-segment concerning the P-wave and ST-segment and QT-interval related to T-wave. Additionally, if the peak areas of the P- and T-waves are deformed and the information is known, the starting point for finding the onset and offset of the P- and T-wave can be changed, as shown in the figure. $v_{P}$ and $v_{T}$ represent the deformation widths of the P- and T-wave and are introduced again in the detection part of the P- and T-point in the next section.

#### 2.2.6. FP Detection of the Q- and S-Wave

The detected R-peak R becomes an important clue to detect the PQ-ST FPs. The Q-point, $Q_{\alpha .L}$, is searched using the following condition while shifting from the location of $R(\mathrm{or}\;R_{L})-1$ to the left (minus direction):(13)$$\begin{array}{l} I(x)=Q_{\gamma }\;\mathrm{if}\;\left\{\begin{array}{l} \gamma \times \mathrm{sign}\left(I^{\prime }((x-1)+n)\right)<0\;\;\mathrm{and}\\ \gamma \times \mathrm{sign}\left(I^{\prime }(x+n)\right)>0 \end{array}\right.\;\;\\ \;\;\;\;\;\;\;\;\;\;\mathrm{for}\;R(\mathrm{or}\;R_{L})-\alpha_{Q}\leq n\leq R(\mathrm{or}\;R_{L})-1 \end{array}$$
where $\alpha_{Q}(=F_{s}\times \mathrm{QRS}/2+\alpha_{\mathrm{QM}})$ is the SR for detecting the Q-point starting from $R(\mathrm{or}\;R_{L})-1$ to the left and $\alpha_{\mathrm{QM}}(=8)$ is the addition margin of the SR for Q-point detection. The subscript γ defines the shape of the Q-point according to the type of R-peak described in Equation (8).

The Q-onset, $Q_{\gamma .L}$, is searched using the following condition while shifting from the $Q_{\gamma }-1$ location to the left (minus direction):(14)$$\begin{array}{l} I(x)=Q_{\gamma .L}\;\mathrm{if}\;\left\{\begin{array}{l} \gamma \times \mathrm{sign}\left(I^{\prime }((x-1)+n)\right)\leq 0\;\;\mathrm{and}\\ \gamma \times \mathrm{sign}\left(I^{\prime }(x+r+n)\right)\geq 0,\;\\ r=0,1 \end{array}\right.\;\;\\ \;\;\;\;\;\;\;\;\;\;\mathrm{for}\;Q_{\gamma }-\alpha_{\mathrm{QL}}\leq n\leq Q_{\gamma }-1 \end{array}$$
where $\alpha_{\mathrm{QL}}(=F_{s}\times \mathrm{QRS})$ represents the SR for detecting the Q-onset starting from $Q_{\gamma }-1$ to the left.

The S-point $S_{\gamma }$ is detected by the following condition while shifting from the $R(\mathrm{or}\;R_{R})+1$ location to the right (plus direction):(15)$$\begin{array}{l} I(x)=S_{\gamma }\;\mathrm{if}\;\left\{\begin{array}{l} \gamma \times \mathrm{sign}\left(I^{\prime }((x-1)+n)\right)<0\;\;\mathrm{and}\\ \gamma \times \mathrm{sign}\left(I^{\prime }(x+n)\right)>0 \end{array}\right.\;\;\\ \;\;\;\;\;\;\;\;\mathrm{for}\;R(\mathrm{or}\;R_{R})+1\leq n\leq R(\mathrm{or}\;R_{R})+\alpha_{S} \end{array}$$
where $\alpha_{S}(=\alpha_{Q})$ is the SR for detecting the Q-point starting from $R(\mathrm{or}\;R_{R})+1$ to the right.

The S-offset $S_{\gamma .R}$ is detected by the following condition while shifting from the $S_{\gamma }+1$ position to the right (plus direction): (16)$$\begin{array}{c} I\left(x\right)=S_{\gamma .R}\;\mathrm{if}\;\left\{\begin{array}{c} \gamma \times \mathrm{sign}\left(I^{\prime }\left(\left(x-1\right)-l-n\right)\right)\leq 0,\;\\ l=0,\;1\;\;\mathrm{and}\\ \gamma \times \mathrm{sign}\left(I^{\prime }\left(x+n\right)\right)\geq 0 \end{array}\right.\;\;\\ \;\;\;\;\;\;\;\;\;\;\;\mathrm{for}\;S_{\gamma }+1\leq n\leq S_{\gamma }+\alpha_{\mathrm{SR}} \end{array}$$
where $\alpha_{\mathrm{SR}}(=\alpha_{\mathrm{QL}})$ is the SR for detecting the S-offset starting from $S_{\gamma }+1$ to the right.

#### 2.2.7. FP Detection of P- and T-Waves

The FPs concerning P- and T-waves are detected on a noise-suppressed predicted background signal to improve the detection accuracy. Based on the heart disease, there is a convex, normal case and a concave, inverted case, and the P- and T-wave-related FPs are composed of the two types, respectively. Based on the SR in Figure 8, the P- and T-points are searched within a given search interval, moving from $Q_{\gamma .L}-1$ to the left (minus direction) and from $S_{\gamma .R}+1$ to the right (plus direction), respectively. First, the symmetric condition for searching for the P-point is as follows:
(17)$$\begin{array}{c} B(x)=P_{\omega }\;\;\mathrm{if}\;\left\{\begin{array}{c} N\;[\omega \times \mathrm{sign}\left(B^{\prime }\left(R+n-l\right)\right)\leq 0,\;\\ \;\;\;\;\;\;l=\left(v_{P}+1\right),\ldots ,\;\left(v_{P}+l_{p}\right),\\ \;\;\;\;\;\;\omega \times \mathrm{sign}\left(B^{\prime }\left(R+n+\left(r-1\right)\right)\right)\geq 0,\;\\ \;\;\;\;\;\;r=\left(v_{P}+1\right),\ldots ,\;\left(v_{P}+r_{p}\right)]\geq \rho_{P}\;\times \left(l_{P}+r_{P}\right)\;) \end{array}\right.\\ \mathrm{for}\;Q_{\gamma .L}-\alpha_{P}\leq n\leq Q_{\gamma .L}-1,\\ \mathrm{where},\;\omega =1\;\;\mathrm{or}\;\;-1,\;\;v_{P}=0,\ldots ,5,\;\;\mathrm{and}\;\;\rho_{P}=0.9 \end{array}$$
where N[⋅] denotes the number of points that satisfy the sign condition in square brackets. ω is the sign converter for detecting normal (ω=+1) or inverted (ω=−1) P- (or T-) points. $v_{P}$ ($v_{T}$ for T-point) defines the number of points adjacent to the point position currently being processed but not used in the above conditional expression for the detection of the P-point (or T-point). $v_{P}$ ($v_{T}$ for T-point) is applied recursively, starting at 0 and until a P-point that satisfies the above condition is detected. It was designed to detect P-waves with a deformed peak. Figure 9 shows the example of the point filter shape for P- and T-point detection at $v_{P}=0\;\;\mathrm{or}\;\;2$, $v_{T}=0\;\;\mathrm{or}\;\;2$, and ω=±1. $l_{P}$ and $r_{P}$ ($l_{T}$ and $r_{T}$ for T-point) denote the left and right surrounding point intervals used directly in the above conditional expression, around the point currently processed for P-wave (T-wave) detection. The left and right wing points used for P-point detection are set as $l_{P}=r_{P}=10-v_{p}$. As illustrated in Figure 8, the lower bound of SR of the P-point is the former T-offset; however, the lower bound value $\alpha_{P}$ was reset to reduce the search range. $\alpha_{P}(=F_{s}\times \mathrm{PR})$ is the SR for detecting the P-point starting from $Q_{\gamma .L}-1$ to the left, and PR(=0.2  sec) is the time of a typical PR-interval. $\rho_{P}(=0.95)$ ($\rho_{T}(=0.95)$ for T-point) denotes the threshold ratio of the left and right point intervals used for P-point (T-point) inspection.

The symmetric condition for searching for the T-point is very similar to the P-point detection procedure of Equation (17) as follows:
(18)$$\begin{array}{c} B(x)=T_{\omega }\;\;\mathrm{if}\;\left\{\begin{array}{c} N\;[\omega \times \mathrm{sign}\left(B^{\prime }\left(R+n-l\right)\right)\leq 0,\;\\ \;\;\;\;\;\;\;l=\left(v_{T}+1\right),\ldots ,\;\left(v_{T}+l_{T}\right),\\ \;\;\;\;\;\;\omega \times \mathrm{sign}\left(B^{\prime }\left(R+n+\left(r-1\right)\right)\right)\geq 0,\;\\ \;\;\;\;\;\;r=\left(v_{T}+1\right),\ldots ,\;\left(v_{T}+r_{T}\right)]\\ \;\;\;\;\;\;\geq \rho_{T}\;\times \left(l_{T}+r_{T}\right)\;) \end{array}\right.\\ \mathrm{for}\;S_{\gamma .R}+1\leq n\leq S_{\gamma .R}+\alpha_{T},\\ \mathrm{where},\;\omega =1\;\;\mathrm{or}\;\;-1,\;\;v_{T}=0,\ldots ,10,\;\;\mathrm{and}\;\;\rho_{T}=0.95 \end{array}$$
where $l_{T}=r_{T}=20-v_{T}$. Furthermore, the upper bound of the SR for the T-point detection is next to the R-peak in Figure 8; however, the upper bound value $\alpha_{T}$ was reset to reduce the search range. $\alpha_{T}(=F_{s}\times (\mathrm{QT}-\mathrm{QRS}))$ is the SR for detecting the T-point starting from $S_{\gamma .R}+1$ to the right, and QT(=0.44  sec) is the typical QT-interval:(19)$$\begin{array}{c} B\left(x\right)=P_{\omega .L},\;\left(\mathrm{or}\;T_{\omega .L}\right)\;\;\mathrm{if}\left\{\begin{array}{c} N[\omega \times \mathrm{sign}\left(B^{\prime }\left(R+n-l\right)\right)\leq 0,\;\;\\ \;\;\;\;\;\;\;l=1,\ldots ,l_{\mathrm{PL}}\;\mathrm{for}\;P_{L}\;\;\left(l=1,\ldots ,l_{\mathrm{TL}}\;\;\mathrm{for}\;T_{L}\right),\;\\ \;\;\;\;\;\;\omega \times \mathrm{sign}\left(B^{\prime }\left(R+n+\left(r-1\right)\right)\right)\geq 0,\;\;\\ \;\;\;\;\;\;\;r=1,\ldots ,r_{\mathrm{PL}}\;\mathrm{for}\;P_{L}\;\;\left(r=1,\ldots ,r_{\mathrm{TL}}\;\;\mathrm{for}\;T_{L}\right)]\\ \;\;\;\;\;\;\;\geq \rho_{\mathrm{PL}}\times \left(l_{\mathrm{PL}}+r_{\mathrm{PL}}\right)\;\mathrm{for}\;P_{L}\;\;\left(\;\rho_{T.L}\times \left(l_{\mathrm{TL}}+r_{\mathrm{TL}}\right)\;\;\mathrm{for}\;T_{L}\right) \end{array}\right.\\ \mathrm{for}\;P_{\omega }-\alpha_{\mathrm{PL}}\leq n\leq \left(P_{\omega }-1\right)+{\tilde{v}}_{P}\;\;\;\mathrm{for}\;\;P_{\omega .L}\;\;\left(S_{\gamma .R}+1\leq n\leq \left(T_{\omega }-1\right)+{\tilde{v}}_{T}\;\;\;\mathrm{for}\;\;T_{\omega .L}\right)\\ \mathrm{where},\;\omega =\pm 1,\;\;l_{\mathrm{PL}}=2,\;r_{\mathrm{PL}}=4,\;\rho_{\mathrm{PL}}=1\;\;\mathrm{for}\;\;P_{L}\;\;\left(\;l_{\mathrm{TL}}=2,\;r_{\mathrm{TL}}=4,\;\;\rho_{\mathrm{TL}}=1\;\;\mathrm{for}\;\;T_{L}\right) \end{array}$$
(20)$$\begin{array}{c} B\left(x\right)=P_{\omega .R},\;\left(\mathrm{or}\;T_{\omega .R}\right)\;\;\mathrm{if}\;\left\{\begin{array}{c} N[\omega \times \mathrm{sign}\left(B^{\prime }\left(R+n-l\right)\right)\leq 0,\;\;\\ \;\;\;\;\;\;l=1,\ldots ,l_{\mathrm{PR}}\;\mathrm{for}\;P_{R}\;\left(\;l=1,\ldots ,l_{\mathrm{TR}}\;\mathrm{for}\;T_{R}\right),\;\\ \omega \times \mathrm{sign}\left(B^{\prime }\left(R+n+\left(r-1\right)\right)\right)\geq 0,\;\;\\ \;\;\;\;\;r=1,\ldots ,r_{\mathrm{PR}}\;\mathrm{for}\;P_{R},\;\;\left(r=1,\ldots ,r_{\mathrm{TR}}\;\mathrm{for}\;T_{R}\right)]\\ \;\;\;\;\;\geq \rho_{\mathrm{PR}}\times \left(l_{\mathrm{PR}}+r_{\mathrm{PR}}\right)\;\mathrm{for}\;P_{R}\;\left(\;\rho_{\mathrm{TR}}\times \left(l_{\mathrm{TR}}+l_{\mathrm{TR}}\right)\;\mathrm{for}\;T_{R}\right) \end{array}\right.\\ \mathrm{for}\;\left(P_{\omega }+1\right)+{\tilde{v}}_{P}\leq n\leq Q_{\gamma .L}-1\;\;\mathrm{for}\;\;P_{\omega .R}\;\;\left(\left(T_{\omega }+1\right)+{\tilde{v}}_{T}\leq n\leq T_{\omega }+\alpha_{\mathrm{TR}}\;\;\mathrm{for}\;\;T_{\omega .R}\right)\\ \mathrm{where},\;\omega =\pm 1,\;l_{\mathrm{PR}}=4,r_{\mathrm{PR}}=2,\;\rho_{\mathrm{PR}}=1\;\;\mathrm{for}\;P_{R}\;\;\left(l_{\mathrm{TR}}=4,\;l_{\mathrm{TR}}=2,\;\rho_{\mathrm{PR}}=1\;\;\mathrm{for}\;T_{R}\right) \end{array}$$

Figure 10 shows the FP detection result of the fragmented P-wave of 118R(200) and 233N(186), the invisible or weak P-wave of 210N(307), 214L(1200), and 220N(303), and the inverted P-wave of 228N(1201). It can be seen that FPs of various abnormally shaped P-waves are detected normally.

As shown in Figure 11, two types of the onset and offset points of the P- and T-waves should also be considered—normal and inverted—depending on ω. The onset and offset point are inflection points at which the slope changes regardless of ω, and the left point section l and the right point section r used for detecting the onset and offset are shown in the figure. The subscripts, PL, PL, TL, and TR, denote the P-onset, P-offset, T-onset, and T-offset searches. The asymmetric condition for detecting the P- and T-onset is defined in Equations (17) and (18). The lower bound of the SR of the P-onset is the preceding T-offset in Figure 8; however, the lower bound value $\alpha_{\mathrm{PL}}=(Q_{\gamma .L}-1)-P_{\omega }$ was reset to reduce the search range based on the morphological symmetry between the P-offset and P-point. ${\tilde{v}}_{P}$ and ${\tilde{v}}_{T}$ are true values obtained by applying $v_{P}$ and $v_{T}$ recursively during the P- and T-point search and are used as the starting points for detecting the P- and T-onset as well as the P- and T-offset. The upper bound of the SR for the T-offset detection is the subsequent R-peak in Figure 8; however, the upper bound value $\alpha_{\mathrm{TR}}=(T_{\omega }-1)-S_{\gamma .R}$ was reset to reduce the SR based on the morphological symmetry between the T-onset and T-point.

In general, the detected P-wave (or T-wave) has a general convex shape, composed of $P_{1}$ (or $T_{1}$), $P_{1.L}$ (or $T_{1.L}$), and $P_{1.R}$ (or $T_{1.R}$) as follows:
(21)$$\left\{\begin{array}{c} P\;\left(\mathrm{or}\;T\right)=P_{-1}\left(\mathrm{or}\;T_{-1}\right),\;\;P_{L}\;\left(\mathrm{or}\;T_{L}\right)=P_{-1,L}\;\left(\mathrm{or}\;T_{-1,L}\right),\;\;P_{R}\;\left(\mathrm{or}\;T_{R}\right)=P_{-1,R}\;\left(\mathrm{or}\;T_{-1,R}\right),\;\;\;\\ \;\;\;\;\;\;\;\mathrm{if}\;\left(\begin{array}{c} \left(P_{-1.R}\;\left(\mathrm{or}\;T_{-1.R}\right)-P_{-1.L}\;\left(\mathrm{or}\;T_{-1.L}\right)\right)>(P_{1.R}\;\left(\mathrm{or}\;T_{1.R}\right)-P_{1.L}\;\left(\mathrm{or}\;T_{1.L}\right)\;\;\\ \mathrm{and}\;\;\Delta_{-1,P}\;\left(\mathrm{or}\;\Delta_{-1,T}\right)>\Delta_{1.P}\;\left(\mathrm{or}\;\Delta_{1.T}\right) \end{array}\right)\\ P\;\left(\mathrm{or}\;T\right)=P_{1}\;\left(\mathrm{or}\;T_{1}\;\right),\;\;P_{L}\;\left(\mathrm{or}\;T_{L}\right)=P_{1,L}\;\left(\mathrm{or}\;T_{1,L}\right),\;\;P_{R}\;\left(\mathrm{or}\;T_{R}\right)=P_{1,R}\;\left(\mathrm{or}\;T_{1,R}\right),\;\;\;\mathrm{otherwise} \end{array}\right.$$
when the certain condition in Equation (21) is satisfied, the P-wave (or T-wave) is assigned to an inverted concave shape and is composed of $P_{-1}$ (or $T_{-1}$), $P_{-1.L}$ (or $T_{-1.L}$), and $P_{-1.R}$ (or $T_{-1.R}$), as in Equation (21). $\Delta_{-1,P}$ and $\Delta_{1,P}$ (or $\Delta_{-1,T}$ and $\Delta_{1,T}$) denote each area estimated by the FPs of the inverted P-wave (or T-wave) and the normal P-wave (or T-wave). The inverted P-wave (or T-wave) corresponds to an inverted triangle while the normal P-wave (or T-wave) is equivalent to a normal triangle. 

Figure 12 shows the FP detection results of the hyperacute T-wave of 106V(497), the biphasic T-wave of 109L(499), the inverted T-wave of 118R(405) and 219x(1200), the camel-hump T-wave of 202N(806), and the deformed T-wave of 201x(608). The FPs of various abnormally shaped T-waves are detected normally.

## 3. Results

### 3.1. PQRST FP Detection Results Using MIT-DB

The effectiveness of the proposed method is analyzed using ECG records with abnormal R-, P-, and T-waves in MIT-DB. Figure 13 shows the PQRST detection results for 106V(497), 108N(6), 118R(159), 201x(608), 202a(1415), 202N(806), 214L(1200), 217f(99), and 233N(186) among the ECG samples of MIT-DB.

### 3.2. Performance Evaluation of R-Peak Detection

To assess the R-peak detection efficiency of the proposed method, several performance indexes are used. The indexes include the false positive (FP) and false negative (FN). The evaluation indices are as follows: sensitivity, Se = TP/(TP + FN), positive prediction, +P[%] = TP(TP + FP), and detection error, DER[%] = (FP + FN)/(TP + FN), respectively. TP represents the total number of accurately detected QRS [52]. The annotated R-point location in each ECG data was used for beat-by-beat comparison with the detected R-peak location based on the WFDB library [53]. If the difference between the annotated position and the detected R-point position is less than 50 ms, the proposed method is considered to have correctly detected the point [54]. Table 2 shows the R-peak detection results by the proposed method for the MIT-DB. The proposed method accomplished Se = 99.83%, +P = 99.82%, and DER = 0.34% for the database.

The proposed R peak detection performance for noises is evaluated in Table 3. Tape 102 is a relatively clean signal with no noise, while date 105 contains large induced noises. For evaluation, the MATLAB function ‘awgn’, which adds white Gaussian noise to the signal, was used for a signal-to-noise ratio (SNR) of 0.5 to 80. The proposed method for noisy signals of 5 dB SNR or higher showed relatively good sensitivity and positive prediction performance, but the performance deteriorated rapidly for noise signals of 1 dB SNR or less.

Table 4 compares the R-peak performance as well as supplemental function for several methods. The proposed method could obtain superior results similar to the existing methods, indicating a sensitivity of 99.83% on 109,510 annotated beats. As shown in Table 4, while all six methods ([5,7,17,19,30], and proposed) indicate the QRS detection technique, only two methods ([11] and proposed) described the QRS-onset and -offset detection technique. Table 4 shows that [11,19,25] accomplished outstanding FP and FN as well as the best DER; thus, they represent fascinating techniques. However, they are not attractive for the detection of the QRS-onset, -offset, and fR-peak. In addition, only the proposed method could detect the fR-peak, which is important for detecting adverse cardiac events. Among the stated methods, only the proposed method can recognize the QRS-onset and -offset as well as the fR-peak. The comparison indicates that the proposed method has relatively high sensitivity, positive prediction as well as low DER by comparison with the previous methods. This demonstrates that the proposed method features superior trade-off between the QRS detection and identification of the fR-peak, QRS-onset, and -offset.

### 3.3. Detection Performance of PQRST Complex Using QT-DB

The QT-DB is used to assess the PQRST detection performance of the proposed method. It includes 105 recordings of 15 min length with 250 Hz, and each recording has a peak and onset and offset annotation information. The annotation for each record was manually performed by two cardiologists and its number reaches 3622 [55,56]. In case the detected S-offset and T-onset are very close (less than about 0.05 s, 13 sample), the detected T-offset was excluded in the evaluation.

Figure 14 shows the PQRST complex detection results of Record Sel41N(5), Sel231N(5), Sel233N(5), Sel301N(5), and Sel808N(5) by the proposed method for QT-DB. The vertical green solid lines show the originally annotated positions in the QT-DB. Figure 14a shows the locations of FPs detected by the proposed method for Sel41N(5), which has an inverted (negative) R-wave, bigger P- and T-waves, and is related to a sudden death. In this connection, the P-peak may be higher than the T-peak, and the interval between the current P-wave and the previous T-wave may be very close. We can see that the proposed method precisely detects the annotated locations. Figure 14b shows the detection result for Sel231N(5), including hyperacute T-waves in a normal sinus rhythm. In the hyperacute T-wave, it is not easy to discriminate between the S-offset and T-onset, so the T-onset has no annotations. Nevertheless, the proposed method detects the T-onset used in HRV. Figure 14c shows the detection result for Sel233N(5), which corresponds to an arrhythmia rhythm and has inverted T-waves, deep S-waves, and a hyperacute T-wave. Because the inverted T-wave appears immediately after the S-wave similar to Sel231N(5), the boundary between the S-offset and T-onset is unclear and the T-onset has no annotation. However, the proposed method estimates the T-onset used on HRV. As a result, we can see that the proposed method detects the peak and offset positions of the inverted T-waves. In addition, the beats with deep S-waves and hyperacute T-waves were not included in the original annotations. Therefore, these beats were excluded from the performance experiment. Figure 14d shows the detection result of Sel301N(5), which corresponds to an ST change rhythm and has a biphasic T-wave. The positive peak was considered as the peak annotated by the cardiologist. The proposed method detects the peak and offset location of T-waves precisely. Figure 14e shows the detection result for Sel808N(5), which is equivalent to a supraventricular rhythm, including inverted T-waves. As in the description of Sel233N(5), since the T-onset is indistinguishable from the S-offset in the inverted T-wave, it was ruled out from the annotations. We can see that the proposed method accurately estimates the original annotation locations for the normal P-wave as well as the inverted T-wave.

Table 5 shows the PQRST complex detection result of the proposed method and existing methods (LPD [33], WT [41], PCGS [44], HFEA [42], MHMM [47]) on 27 Records (19 Records except for MHMM from the original paper) from the QT-DB. However, the result of TLBLDT was obtained by the LU database [48]. Two-standard-deviation tolerances have been included in the last row [57]. The mean (m) and standard deviation (s) between the detected positions by respective methods and the originally annotated positions are also listed. The root mean square error ($\mathrm{RMSE}=\sqrt{\left(m^{2}+s^{2}\right)}$) is used for quantitative evaluation of the respective methods [47]. The blue and green numbers in the table represent the smallest and the second smallest RMSE of the location difference of the FPs. The proposed method has the smallest RMSE value in the P-onset, P-offset, and T-wave, and the second lowest RMSE value in the P-peak, R-peak, and QRS-onset among all methods. The standard deviation of the proposed method is lower than the CSE tolerance in the P-offset, QRS-offset, and T-offset, and is close to the CSE tolerance in the P-onset and QRS-onset. Additionally, the proposed method also satisfies the criteria limit of the CSE tolerance for the P-offset with LPD and MHMM. Unfortunately, no method yet satisfies the severe criteria of the QRS-onset. The mean and standard deviation of all FPs are evaluated for LPD, WT, PCGS, HFEA, MHMM, TLBLDT, and the proposed method as 2.9 ± 13.5, 1.6 ± 12.7, 2.9 ± 15.1, −29 ± 17.4, −1.0 ± 8.5, 0.4 ± 7.4, and 0.4 ± 7.4 ms, respectively. The RMSE values across all FPs for the above-mentioned methods are evaluated to be 15.3, 13.1, 15.7, 19.8, 8.9, 9.5, and 8.3, respectively.

### 3.4. HRV Analysis for MIT-DB

The heart is continually regulated by the autonomic nervous system [58]. HRV denotes the time variation between successive heartbeats [58,59]. HRV can estimate the sympathovagal balance through time- and frequency-domain methods [58]. In the time domain, HRV is assessed by the variability of heartbeats observed in the range of 5 min to 24 h [60]. These methods are calculated based on the beat-to-beat (or NN) intervals. The statistical time-domain measures (or metrics) include the mean HR, HR standard deviation (std.), RR-mean, NN50, pNN50, RMSSD, HRV triangular index (HTI), and TINN, which are listed in Table 6 [59,61,62,63].

For the HR, variation in certain frequency bands is a very useful method to assess the sympathetic-parasympathetic balance of the cardiovascular system. Additionally, the high frequency band (HF) is also denoted as respiratory arrhythmia because it is associated with HRV associated with the breathing cycle and is controlled by the activity of the parasympathetic nervous system (PNS). The low frequency band (LF) is regulated by both the sympathetic nervous system (SNS) and PNS. The very low frequency band (VLF) is a variable dependent on the Lenin-Angiotensin system and is also affected in both SNS and PNS. A reduction in the ultra-low-frequency and VLF in 24-h ECG is the strongest predictor of acute heart attack and death from arrhythmia. Spectrum analysis is a useful method for evaluating the effects of the SNS and PNS at the same time. However, because individual deviations vary, it is difficult to obtain a normal number that can be used as a test tool. Therefore, it should always be investigated to see if it has decreased or increased statistically compared with the control group. The reduction in HRV mentioned above is a very powerful prognostic factor in all diseases with cardiovascular abnormalities.

Figure 15 shows the histogram of RRIs, spectral power, ST-segment, QT-interval, and PR-interval for 5 min of Record 106V(497), 108N(6), 118R(159), 201x(608), 202a(1415), and 217f(99) in MIT-DB.

The ST-segment progresses from the S-offset to the T-onset of the following T-wave. This is the sluggish depolarization term of the ventricles after the contractions of the two ventricles [64]. The baseline of the ST-segment typically is near the isoelectric line (IL) by height. The IL corresponds to the baseline of the whole ECG wave, typically located at 0 mV. Abnormal cardiac activities make the baseline of the ST-segment usually rise or fall from the IL. Elevation of the ST-segment represents myocardial infraction. On the other hand, depression of the ST-segment is typically relevant to hypokalemia [65].

The typical term of the ST-segment corresponds to 0.005 to 0.15 s. It can be seen that all test signals have a normal range of ST-segments. Furthermore, 106V(497) (female, age 24) shows a relatively constant ST-segment of about 0.1 s or less. Moreover, 108N(6) has a somewhat irregular ST-segment, which is close to zero up to 100 s due to the effects of asymmetrically peaked (or hyperacute) T-waves. However, after 100 s, the normal T-waves are observed, but an unstable ST-segment distribution is observed due to the delay of the T-wave (ST-segment above 0.15 s) and noise (ST-segment close to zero). Additionally, 118R(159) has RBBB’s inverse T-waves but shows a very constant ST-segment distribution near zero and 201x(608) (male, age 68) shows a slightly higher ST-segment of 0.2 s due to the slowly rising T-waves up to 510 s and shows an ST-segment close to 0 s because of the deformation of the T-waves by the annotation “x” (Non-conducted P-wave, blocked APB) and the annotation “a” (Aberrared atrial premature) [66]. However, from 510 s, it frequently shows ST-segments close to zero due to the frequent appearance of the annotation “V” (PVC). 202a(1415) (male, age 68) basically exhibits a relatively normal range of ST-segments but shows an unstable ST-segment distribution due to the gentle T-waves or T-wave delay and biphasic T-waves. 217f(99) shows an ST-segment distribution close to zero due to asymmetrically peaked (or hyperacute) T-waves. We know that the ST-segment is not easy to distinguish whether the detected T-wave is normal, an hyperacute T-wave, or a biphasic T-wave with a large falling curve.

The PR-interval represents the term from the P-onset to the Q-onset. It indicates the conduction by means of the AV node. In case the PR-interval is longer than 0.2 s, it can be considered that a first-degree heart block exists [67]. The prolonged PR-interval arouses delayed conduction by the AV node. Furthermore, in case the PR-interval is less than 0.12 s, it is considered to manifest a pre-excitation or AV nodal rhythm.

The typical term of the PR-interval corresponds to the range of 0.12 to 0.2 s. 106V(497), 108N(6), 201x(608), and 217f(99) show a normal distribution, whereas 118R(159) and 202a(1415) show a somewhat unstable distribution. 106V(497) (female, age 24) has normal PR-intervals overall due to very clean P-waves. 108N(6) (female, age 87) also has a normal PR-interval and has a PR-interval of about 0.3 s because of the P-waves, which gradually increases in height. 118R(159) (male, age 69) has a notch (left atrial enlargement symptom) at the P-peaks, which is due to longer atrial depolarization times than normal. Therefore, the signal has a rather long PR-interval (about 0.37 secs). At 201 × 608 (male, age 68), the onset of AF with aberrated beats occurred at approximately 472 s. The ventricular fibrillation (AFIB) is the most severe from 480 to 520 s, and during this time, the P-waves are so weak that they cannot be detected; thus, the signal shows very low PR-intervals. After 550, 650, and 700 s, the annotation “V” (PVC) occurs frequently with little visible P-waves (“x” (paced) is very rarely present), resulting in very low PR-intervals. At 202a (1415) (male, age 68), AF with aberrated beats occurred at 1333 s, and then, atrial flutter with 2: 1 conduction occurred at 1558 s. This caused a momentary PR-interval drop near 1550 s. For the entire signal, the P-wave was very weak and accompanied by noise, which shows that the PR-interval is unstable. 217f (99) (male, age 65) shows relatively constant PR-intervals between 0.2 and 0.3 s, although the P-waves are very weak due to “/” (paced). However, because of the intermittent “f” (Fusion of paced and normal) or “V” (PVC), there are some intervals without P-waves; thus, the PR-interval with zero is partially visible.

The QT-interval represents the Q-onset to the T-offset. Specifically, it indicates the time from ventricle depolarization to ventricle repolarization. The unusually shortened or prolonged interval increases the danger of revelation of ventricular arrhythmia [68]. In special cases, a prolonged QT-interval can cause a life-threatening ventricular tachycardia.

The typical term of the QT-interval corresponds to the range of 0.32 to 0.44 s. 106V(497) has a very constant QT-interval of 0.4 s, except for the long QT-interval (>0.6 s), due to the very high and wide T-waves of “V” (PVC) that appear around 450 s and after 710 s. At 108N(6), QT-intervals (more than 0.6 secs) beyond the normal range are observed because of the low but slowly increasing and then decreasing T-waves in the interval of less than 100 s and more than 160 s. In addition, the QT-interval was miscalculated because of severe noise near 150 and 250 s. 118R(159) has somewhat constant QT-intervals of 0.4 s because of clean inverse T-waves with a very constant ST-segment of 0.1 to 0.2 s. In the case of 201x(608), AFIB occurred at approximately 500 s due to the onset of AF with aberrated beats, which occurred at 472 s. Therefore, P- and T-waves were hardly seen and were not detected. Furthermore, the QT-interval near 500 s was momentarily reduced to 0.2 s. 202a(1415) has a QT-interval in the normal range of 0.4 s overall but exhibits a somewhat variable QT-interval distribution due to the gentle T-wave, T-wave delay, and biphasic T-wave as mentioned in the ST-segment experiments above. 217f(99) has a rather high QT-interval of approximately 0.6 s. 217f(99) has intermittently increasing QT-intervals due to a very tall asymmetrically peaked (or hyperacute) T-waves.

Table 7 shows the HRV analysis result of the test signals in MIT-DB. Bradycardia represents the HR of 45 beats/min or less while tachycardia denotes the HR of 100 beats/min or more at rest [69]. All the test signals corresponded to a regular HR. 201x(608) with high RR std. and HR std. exhibited irregular RRI and HR, whereas 202a(1415) with a high HR std. has an irregular HR but showed a regular RRI. In fact, the high RR std. and HR of 201x(608) are due to fast and irregular “N” beats, “V”, “A”, and “x” (actually AFIB), and the irregular HR of 202a(1415) is due to irregular “N” beats per minute (actually AFIB). NN50, pNN50, and RMSSD indicate short-term cardiac variability and are associated with parasympathetic nerve activity. Additionally, those variables are deeply related to AF and sudden death [70]. 201x(608) shows the highest RMSSD, NN50, and pNN50 among the tested signals and was actually annotated with AFIB at about 8:00. The patient was prescribed hypertension medication [71]. 202a (1415) had a relatively high RMSNN but relatively high NN50 and pNN50 values, and was diagnosed with AFIB annotated at about 26:20; this patient was also prescribed medication for hypertension [72]. In a previous study, when HTI > 20.42, the pattern was considered arrhythmic [73]. As a criterion, 202a(1415) (HTI = 18.9) close to the threshold is classified as arrhythmia. On the other hand, since the remaining test signals show an HTI of 20 or less, those signals can be regarded as normal. However, specifically, 201x(608) had a very low HTI (2.693) but had AFIB. It can be seen that it is difficult to determine whether a signal is normal or not by means of only HTI. 201x(608) with TINN <15 can be estimated to have severely decreased HRV [74].

HRV variables with a lower value indicate greater cardiac risk. VLF power is more deeply related to all-cause mortality compared to LF and HF power [73]. Low VLF power has been researched to be related to arrhythmic death and is also associated with high inflammation [73,74]. The VLF rhythm appears to be influenced by the intrinsic nervous system of the heart, and the SNS affects the amplitude and frequency of its variations.

The power of most test signals is converged in DC to constant RRIs. Because the power spectral density only deals with the frequency of the RRIs, it is not directly related to BPM. In other words, in case the BPM is constant regardless of whether it is bradycardia (or tachycardia), it may show a high VLF value. It can be seen that the 108N(6) has a lower than average BPM than other test signals but have as much power as other signals in the VLF band. 201x(608) has more HF power compared to other test signals, which indicates that the RRI is unstable. In addition, in fact, 201x(608) contains numerous “V” and some “a” and “j” annotation beats with AFIB. 202a(1415) also contains AFIB and atrial flutter (AFL); however, no difference was observed in the HF power. As explained above, while LF works through sympathetic nerves, HF is affected by parasympathetic nerves [73]. A low LF/HF, such as those for 106V(497), and 118R(159), denotes when to conserve energy, whereas a high LF/HF, such as those 108N(6), 202a(1415), and 217f(99), exhibits fight-or-flight behavior.

## 4. Conclusions

This study presents a novel PQRST complex detector using edge preservation and noise suppression characteristics of 1DBF and the temporal characteristics of FPs. All FPs are detected by a predefined detection order and mapping method of absent FPs. The evaluation of the proposed method was proved using HRV analysis using the detected FPs as well as the detection comparison of the FPs. The detection of PQRST FPs is also very important for the classification of various arrhythmias. The current medical and healthcare devices are evolving beyond binary classification of normal and abnormal to ECG signals to predict specific heart disease. The current HRV is analyzed by the length of time between each FP; however, the amplitude of each FP must also be reflected in the HRV for detailed classification of arrhythmia and pathological prediction of the heart. In the future, more accurate PQRST complex detectors and amplitude reflecting novel HRVs must be developed.

## Figures and Tables

**Figure 1 ijerph-18-10792-f001:**
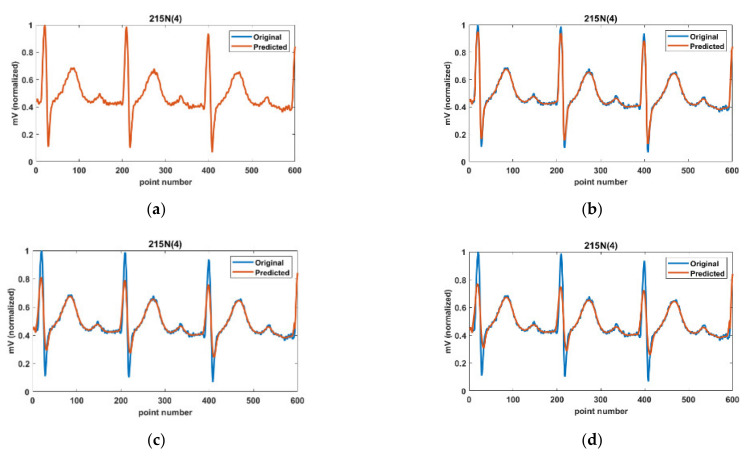
Impact of standard deviation; (**a**)
$\sigma_{d}=0.1$,
$\sigma_{r}=0.1$, (**b**)
$\sigma_{d}=5$,
$\sigma_{r}=0.1$, (**c**)
$\sigma_{d}=5$,
$\sigma_{r}=1$, and (**d**)
$\sigma_{d}=10$,
$\sigma_{r}=1$
in 215 N(4).

**Figure 2 ijerph-18-10792-f002:**
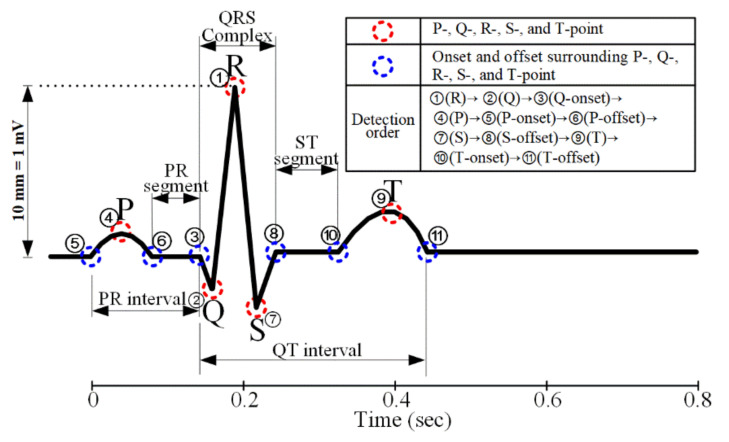
FPs concerning the PQRST complex and detection order.

**Figure 3 ijerph-18-10792-f003:**
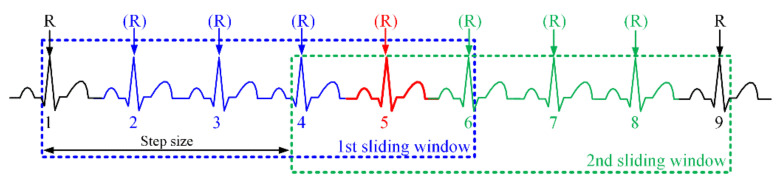
An example of seamless connection of successive SWs for real time.

**Figure 4 ijerph-18-10792-f004:**
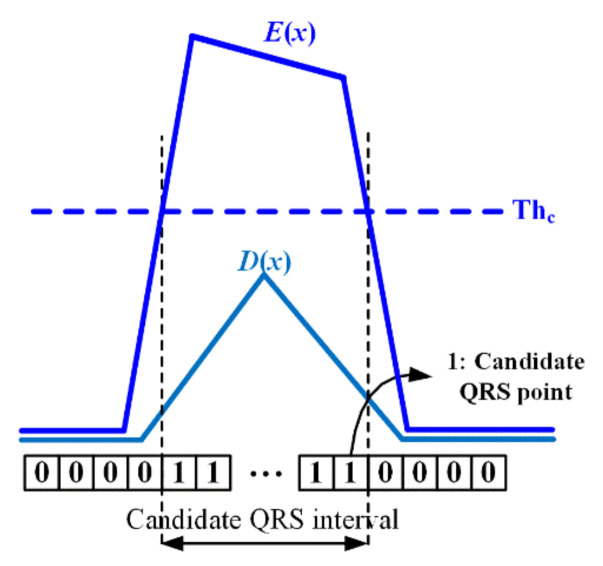
Candidate-R section in the enhanced residual signal.

**Figure 5 ijerph-18-10792-f005:**
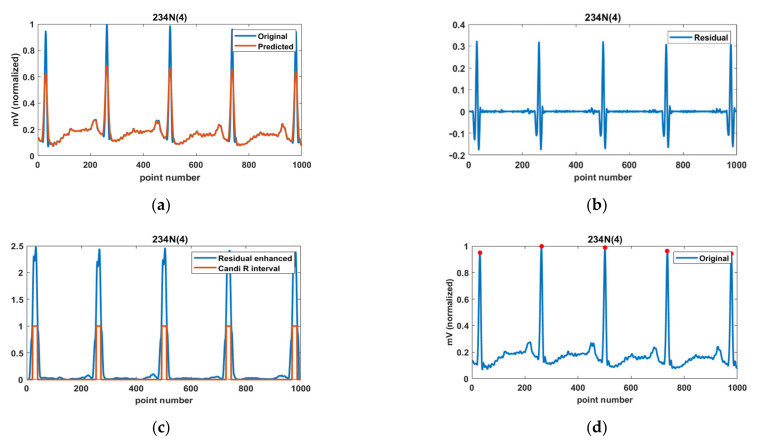
Detection procedure of the R-peak; (**a**) predicted background signal, (**b**) residual (difference) signal, (**c**) candidate-R interval, and (**d**) detected R-peak.

**Figure 6 ijerph-18-10792-f006:**
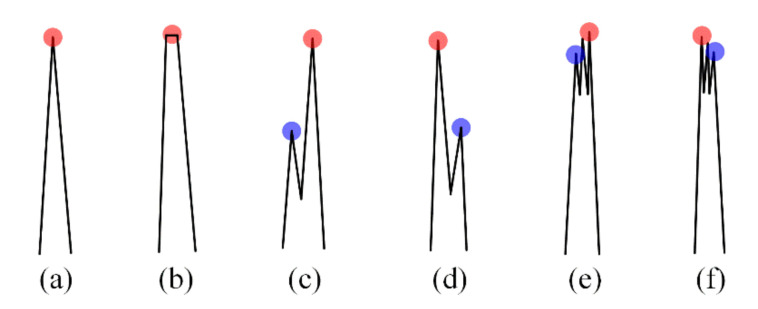
The various R-peak shape; (**a**) normal R, (**b**) flat R, (**c**) fR-L, (**d**) fR-R, (**e**) fR-L2, and (**f**) fR-R2.

**Figure 7 ijerph-18-10792-f007:**
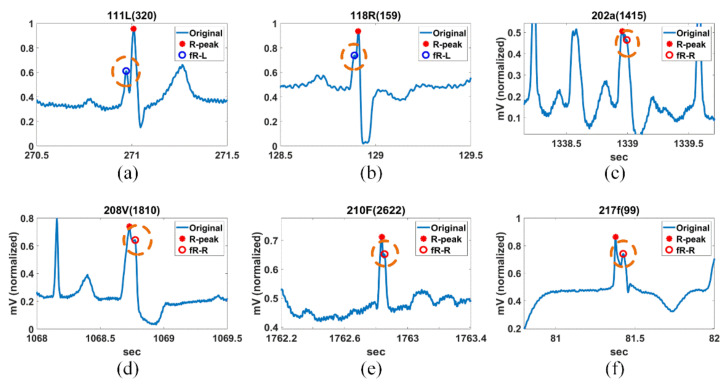
Detection of the R-peak and additional R-peak for (**a**) 111L(320), (**b**) 118R(159), (**c**) 202a(1415), (**d**) 208V(1810), (**e**) 210F(2622), and (**f**) 217f(99) in MIT-DB.

**Figure 8 ijerph-18-10792-f008:**
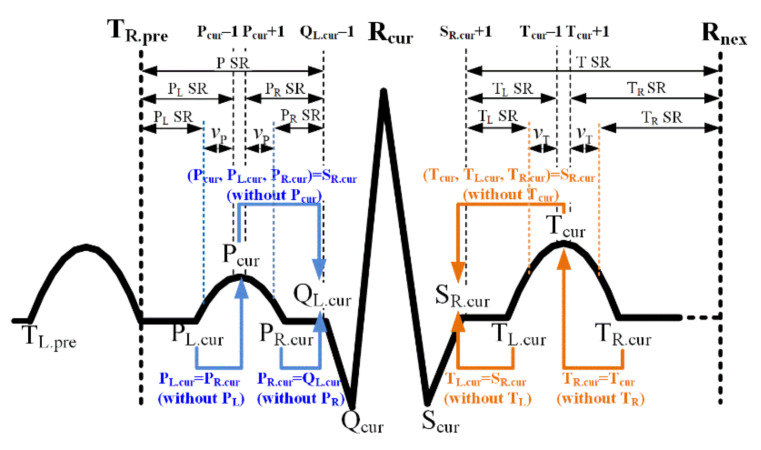
The SR for FP detection of P- and T-wave; Subscripts L, R, pre, cur, pre, and nex denote onset, offset, previous, current, next.

**Figure 9 ijerph-18-10792-f009:**
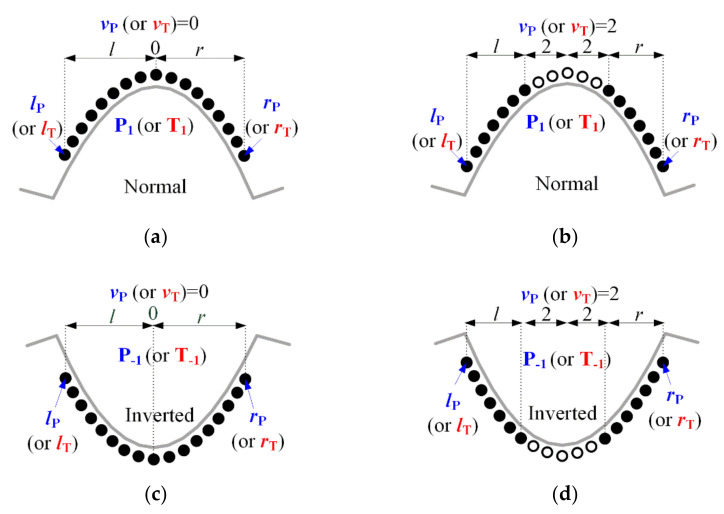
Symmetric point filters for normal and abnormal P- and T-point detection; (**a**) normal type (ω=1, $v_{P}=v_{T}=0$), (**b**) deformed normal type (ω=1, $v_{P}=v_{T}=2$), (**c**) inverted type (ω=−1, $v_{P}=v_{T}=0$), and (**d**) deformed inverted type (ω=−1,$v_{P}=v_{T}=2$).

**Figure 10 ijerph-18-10792-f010:**
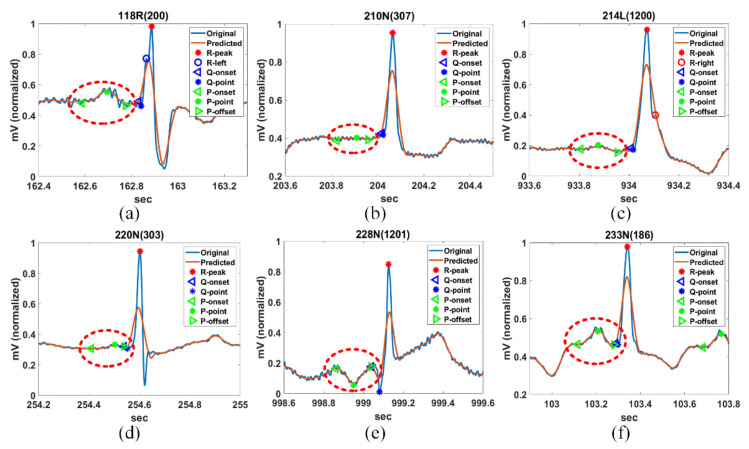
Detection of abnormal P-waves for (**a**) 118R(200), (**b**) 210N(307), (**c**) 214L(1200), (**d**) 220N(303), (**e**) 228N(1201), and (**f**) 233N(186) in MIT-DB.

**Figure 11 ijerph-18-10792-f011:**
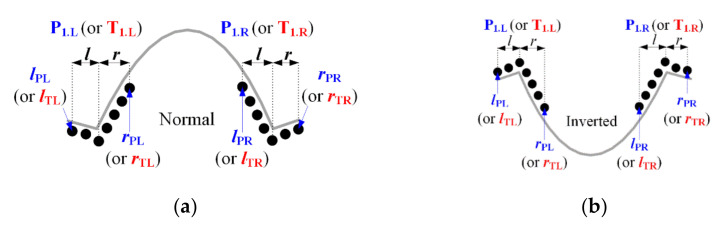
Asymmetric point filters for onset and offset detection of the P- and T-wave; (**a**) normal type and (**b**) inverted type.

**Figure 12 ijerph-18-10792-f012:**
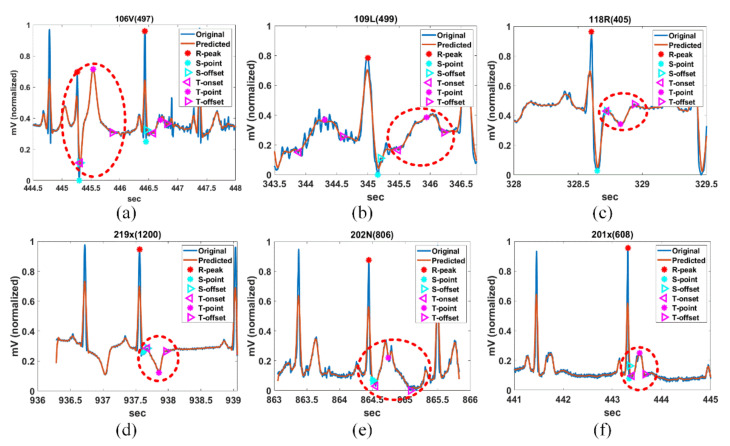
Detection of abnormal T-waves for (**a**) 106V(497), (**b**) 109L(499), (**c**) 118R(405), (**d**) 201x(608), (**e**) 202N(806), and (**f**) 201x(608) in MIT-DB.

**Figure 13 ijerph-18-10792-f013:**
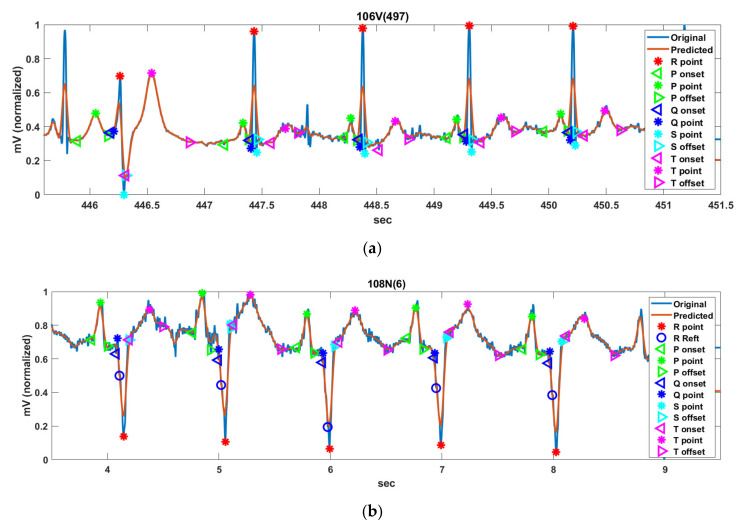
Detection result of the PQRST FPs for Record (**a**) 106V(497), (**b**) 108N(6), (**c**) 118R(159), (**d**) 201x(608), (**e**) 202a(1415), (**f**) 202N(806), (**g**) 214L(1200), (**h**) 217f(99), and (**i**) 233N(186) sample in MIT-DB.

**Figure 14 ijerph-18-10792-f014:**
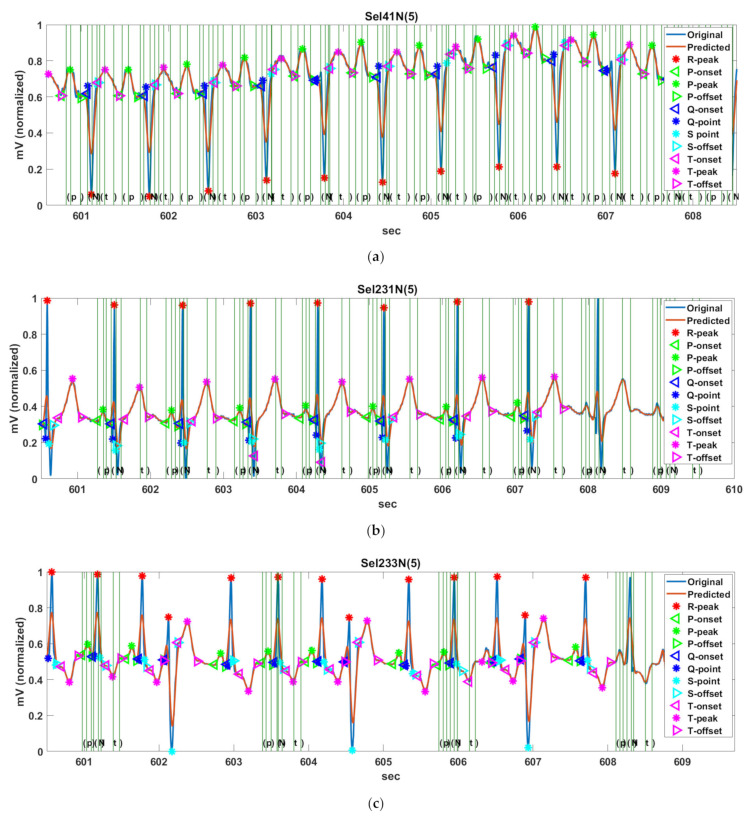
PQRST complex detection results of (**a**) Sel41N(5), (**b**) Sel231N(5), (**c**) Sel233N(5), (**d**) Sel301N(5), and (**e**) Sel808N(5) by the proposed method in QT-DB.

**Figure 15 ijerph-18-10792-f015:**
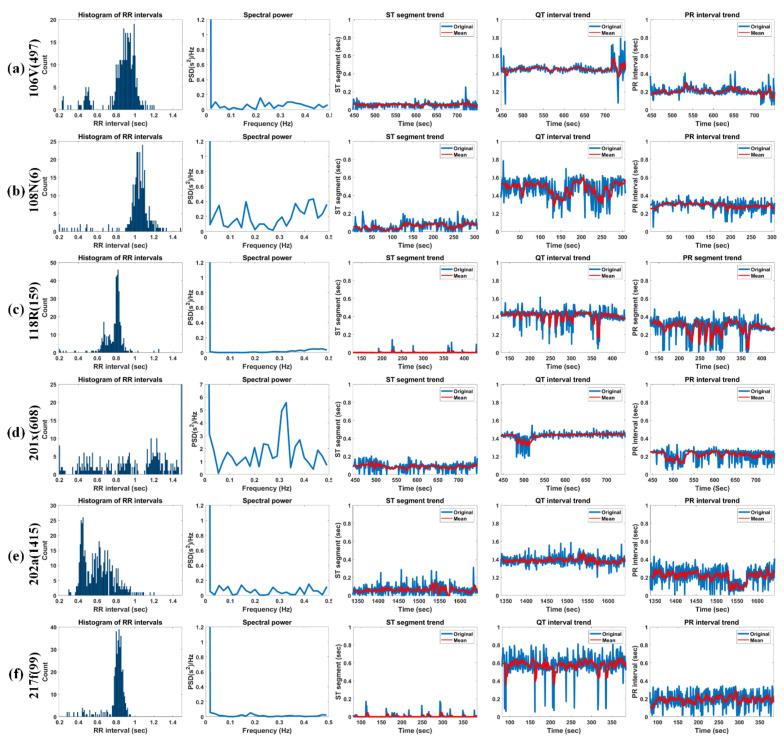
HR trend, RRI trend, histogram of RRIs, spectral power, ST-segment, QT-interval, and PR-interval for 5 min length of Record (**a**) 106V(497), (**b**) 108N(6), (**c**) 118R(159), (**d**) 201x(608), (**e**) 202a(1415), and (**f**) 217f(99) in MIT-DB.

**Table 1 ijerph-18-10792-t001:** Five kinds of heartbeat annotated in MIT-DB.

N (Non-Ectopic)	S (Supraventricular Ectopic)	V (Ventricular Ectopic)	F (Fusion)	Q (Unknown)
- Normal (N) - Left bundle branch block (L) - Right bundle branch block (R) - Atrial escape (e) - Nodal (junctional) escape (j)	- Atrial premature (A) - Aberrated atrial premature (a) - Nodal (junctional) premature (J) - Supra-ventricular premature (S)	- Premature ventricular contraction (V) - Ventricular escape (E)	- Fusion of ventricular and normal (F)	- Paced (/) - Fusion of paced and—normal (f) - Unclassified (Q)

**Table 2 ijerph-18-10792-t002:** R-peak detection performance of the proposed method for the MIT-DB.

Tape	Total	FN	FP	Se [%]	+P [%]	DER [%]	Tape	Total	FN	FP	Se [%]	+P [%]	DER [%]
100	2273	0	0	100	100	0	201	1963	5	2	99.75	99.90	0.36
101	1865	2	1	99.89	99.95	0.16	202	2136	3	3	99.86	99.86	0.28
102	2187	0	0	100	100	0	203	2980	13	15	99.56	99.50	0.94
103	2084	0	0	100	100	0	205	2656	6	3	99.77	99.92	0.30
104	2229	4	14	99.82	99.37	0.81	207	1862	7	8	99.62	99.57	0.81
105	2572	17	12	99.34	99.53	1.13	208	2955	17	9	99.42	99.69	0.88
106	2027	11	16	99.46	99.21	1.33	209	3004	5	3	99.83	99.90	0.27
107	2137	2	3	99.91	99.86	0.23	210	2650	12	9	99.55	99.66	0.79
108	1774	13	23	99.27	98.71	2.03	212	2748	1	5	99.96	99.82	0.22
109	2532	3	0	99.88	100	0.12	213	3251	2	3	99.94	99.91	0.15
111	2124	1	2	99.95	99.91	0.14	214	2265	0	1	100	99.96	0.04
112	2539	0	1	100	99.96	0.04	215	3363	3	3	99.91	99.91	0.18
113	1795	2	1	99.89	99.94	0.17	217	2209	5	6	99.77	99.73	0.50
114	1879	3	4	99.84	99.79	0.37	219	2154	1	3	99.95	99.86	0.19
115	1953	0	0	100	100	0	220	2048	1	0	99.95	100	0.05
116	2412	13	4	99.46	99.83	0.70	221	2427	3	2	99.88	99.92	0.19
117	1535	0	2	100	99.93	0.07	222	2483	2	2	99.92	99.92	0.16
118	2278	2	1	99.91	99.96	0.13	223	2605	3	2	99.88	99.92	0.19
119	1987	1	0	99.95	100	0.05	228	2053	4	13	99.81	99.37	0.83
121	1863	2	0	99.89	100	0.11	230	2256	1	4	99.96	99.82	0.22
122	2476	0	0	100	100	0	231	1571	1	2	99.94	99.87	0.19
123	1518	2	1	99.87	99.93	0.20	232	1780	2	3	99.89	99.89	0.22
124	1619	0	0	100	100	0	233	3079	4	1	99.87	99.97	0.16
200	2601	2	9	99.92	99.65	0.42	234	2753	3	0	99.89	100	0.11
Total								109,510	184	193	99.83	99.82	0.34

**Table 3 ijerph-18-10792-t003:** R-peak detection performance of the proposed method for SNR change.

SNR	Tape	Total	FN	FP	Se [%]	+P [%]	Tape	Total	FN	FP	Se [%]	+P [%]
0.5	102	2187	781	732	64.29	65.76	105	2572	887	789	65.51	68.11
1			551	455	74.81	78.24			651	639	74.69	75.04
5			259	232	88.16	89.26			389	352	84.88	86.11
10			121	115	94.47	94.73			189	153	92.65	93.97
15			63	55	97.12	97.48			112	78	95.65	96.93
20			33	29	98.49	98.67			61	49	97.63	98.09
40			11	7	99.50	99.68			23	16	99.11	99.38
60			3	0	99.86	100			19	13	99.26	99.49
80			0	0	100	100			17	12	99.34	99.53

**Table 4 ijerph-18-10792-t004:** Comparison of R-peak detection performance and supplemental function for several methods.

Method	Onset and Offset Detection Detection of QRS	Number of R Classification	fR Detection	Se [%]	+P [%]	DER [%]	TB	TP	FN	FP
Pan and Tompkins [5]	N (R, QRS)	*	N	99.76	99.56	0.68	116,137	115,860	277	507
Hamilton and Tompkins [7]	N (R, QRS)	*	N	99.69	99.77	0.54	109,267	108,927	340	248
Farashi [21]	N (Only R)	2	N	99.75	99.85	0.40	109,965	109,692	273	163
Phukpattaranont [22]	N (Only R)	2	N	99.82	99.81	0.38	109,483	109,281	202	210
Manikandan and Soman [23]	N (Only R)	1	N	99.93	99.87	0.20	109,496	109,417	79	140
Merah et al. [11]	Y	1	N	99.84	99.88	0.28	109,494	109,316	178	126
Cristov [16] (algorithm 2)	N (Only R)	*	N	99.78	99.78	0.44	110,050	109,615	240	239
Karimipour and Homaeinezhad [17]	N (R, QRS)	*	N	99.81	99.70	0.43	116,137	115,945	192	308
Dohare et al. [30]	N (R, QRS)	1	N	99.21	99.34	1.45	109,966	109,096	870	728
Yazdani and Vesin [19]	N (R, QRS)	2	N	99.87	99.90	0.22	109,494	109,357	137	108
Castells-Rufas and Carrabina [9]	N (Only R)	*	N	99.43	99.67	0.88	109,494	108,880	614	353
Jinkwon Kim and Hangsik Shin [25]	N (Only R)	*	N	99.90	99.91	0.19	109,494	109,357	107	97
DS Benitez et al. [13]	N (Only R)	1	N	99.81	99.83	0.36	N/A	N/A	203	187
Proposed method	Y	4	Y	99.83	99.82	0.34	109,510	109,326	184	193

* Only QRS complexes detection.

**Table 5 ijerph-18-10792-t005:** Performance comparison of PQRST complex detection from QT-DB (N/A: Not applicable).

Methods	Fiducial Points								
	P_onset_	P_peak_	P_offset_	QRS_onset_	R_peak_	QRS_offset_	T_onset_	T_peak_	T_offset_
LPD [33]	14 ± 13.3 (19.3)	4.8 ± 10.6 (11.6)	−0.1 ± 12.3 (12.3)	−3.6 ± 8.6 (9.3)	N/A (N/A)	−1.1 ± 8.3 (8.4)	N/A (N/A)	−7.2 ± 14.3 (16.0)	13.5 ± 27.0 (30.2)
WT [41]	2.0 ± 14.8 (14.9)	3.6 ± 13.2 (13.7)	1.9 ± 12.8 (13)	4.6 ± 7.7 (9)	N/A (N/A)	0.8 ± 8.7 (8.7)	N/A (N/A)	0.2 ± 13.9 (13.9)	−1.6 ± 18.1 (18.2)
PCGS [44]	3.7 ± 17.3 (17.7)	4.1 ± 8.6 (9.5)	−3.1 ± 15.1 (15.4)	N/A (N/A)	N/A (N/A)	N/A (N/A)	7.1 ± 18.5 (19.8)	1.3 ± 10.5 (10.6)	4.3 ± 20.8 (21.2)
HFEA [42]	−6.3 ± 12.5 (14.0)	5.0 ± 9.5 (10.7)	3.1 ± 16.0 (16.3)	3.7 ± 7.8 (8.6)	3.8 ± 9.8 (10.5)	12.1 ± 16.6 (20.6)	−15.8 ± 34.0 (37.5)	−15.3 ± 29.3 (33.0)	−16.6 ± 20.8 (26.6)
MHMM [47]	4.0 ± 12.0 (12.5)	0.2 ± 3.5 (3.4)	−3.0 ± 11.0 (12.2)	−5.0 ± 10.0 (11)	0 ± 0.2 (0.2)	1.5 ± 11.5 (11.6)	N/A (N/A)	−0.4 ± 5.6 (5.6)	−5.0 ± 14.0 (14.7)
TLBLDT [48]	2.2 ± 7.4 (7.7)	−0.76 ± 5.5(5.6)	−6.5 ± 10.7(12.5)	15.4 ± 14.6(21.2)	−0.14 ± 3.4(3.4)	−3.8 ± 13.6(14.1)	−1.3 ± 8.8(8.9)	−0.5 ± 5.5(5.5)	−1.2 ± 6.8(6.9)
Proposed	3.8 ± 10.9 (11.5)	1.8 ± 3.3 (3.8)	3.2 ± 9.6 (10.1)	−3.9 ± 8.2 (9.1)	0.2 ± 0.3 (0.4)	−3.1 ± 8.7 (9.2)	8.3 ± 10.7 (13.5)	0.7 ± 4.6 (4.7)	−7.2 ± 10.2 (12.5)
Tolerances (2s_CSE_) [57]	10.2	-	12.7	6.5	-	11.6	-	-	30.6

**Table 6 ijerph-18-10792-t006:** Indices concerning time- and frequency-domain methods of HRV.

Variable			Units	Meaning
Time domain	Statistical	Mean heart rate	[s]	-
		heart rate std.	[s]	-
		RR mean	[ms] or [s]	Average RR interval in the window of measurement
		NN50	[count]	The number of adjacent RR intervals that varied by more than 50 ms
		pNN50	[%]	The percentage of adjacent RR intervals that varied by more than 50 ms
		rMSSD	[ms]	Root mean square of the difference between the coupling intervals of adjacent RR intervals
	Geometric	HRV triangular index (HTI)	-	A measure in which the length of RR intervals serves as the x-axis of the plot and the number of each RR interval length serves as the y-axis
		TINN	-	Triangular interpolation, the baseline width of the distribution measured as a base of a triangle
Frequency domain		VLF power	[ms^2^]	Power from very low frequency (0 Hz~0.04 Hz)
		LF power	[ms^2^]	Power from low frequency (0.04 Hz~0.15 Hz)
		HF power	[ms^2^]	Power from high frequency (0.15 Hz~0.40 Hz)
		VLF	[%]	(VLF Power/Total Power) × 100
		LF	[%]	(LF Power/Total Power) × 100
		HF	[%]	(HF Power/Total Power) × 100
		LF norm	n.u.	LF power in normalized units: LF/(Total Power—VLF) × 100
		HF norm	n.u.	HF power in normalized units: HF/(Total Power—VLF) × 100
		LF/HF	-	Sympathovagal balance

**Table 7 ijerph-18-10792-t007:** HRV analysis for test records of MIT-DB.

Variables	106V(497)	108N(6)	118R(159)	201x(608)	202a(1415)	217f(99)
RR mean	0.872	1.028	0.786	0.979	0.617	0.806
RR std.	0.171	0.177	0.110	0.399	0.159	0.111
Heart rate mean	66.75	58.8	77	63.75	97.8	74.8
Heart rate std.	2.217	2.388	4.743	11.117	15.238	1.483
RMSSD	0.201	0.196	0.148	0.566	0.178	0.126
NN50	75	79	46	159	165	55
pNN50	21.676	26.963	11.979	51.961	33.605	14.706
HTI	18.263	12.25	8.370	2.693	18.923	9.615
TINN	32	27	31	7	11	21
VLF power	8.073 × 10^5^	1.01 × 10^6^	6.75 × 10^5^	1.07 × 10^6^	5.02 × 10^5^	6.68 × 10^5^
LF power	5.18 × 10^3^	2.63 × 10^4^	2.72 × 10^2^	1.64 × 10^5^	9.93 × 10^3^	2.10 × 10^3^
HF power	1.78 × 10^4^	4.98 × 10^4^	3.92 × 10^3^	5.63 × 10^5^	1.56 × 10^4^	3.65 × 10^3^
VLF	97.270	92.697	98.923	59.484	95.471	99.222
LF	0.625	2.418	0.040	9.106	1.889	0.313
HF	2.155	4.574	0.575	31.275	2.971	0.543
LF norm	22.897	33.102	3.697	22.474	41.698	40.215
HF norm	78.933	62.625	53.371	77.192	65.603	69.822
LF/HF	0.290	0.529	0.069	0.291	0.636	0.576

## Data Availability

Not applicable.

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
