# Peer review of "Electrocardiogram Fiducial Point Detector Using a Bilateral Filter and Symmetrical Point-Filter Structure"

_ijerph, 2021, doi:10.3390/ijerph182010792_

Round 1
Reviewer 1 Report
I suggest the authors improve the quality of the tables and figures so that they can be properly interpreted. Additionally, I suggest that Figure 19 be presented in different figures, because what is written in it is not appreciated.
I suggest to the authors that the text on lines 168-170 be placed on line 166, so that table 1 is introduced and does not appear right after subheading 2.2.3.
Similarly, I suggest that figures be introduced first in the text and then included. For example, figure 4 I would introduce on line 191, figure 5 on line 223, figure 6 on line 236, figure 7 on line 271, figure 9 on line 322, figure 14 on line 453.
Author Response
I suggest the authors improve the quality of the tables and figures so that they can be properly interpreted. Additionally, I suggest that Figure 19 be presented in different figures, because what is written in it is not appreciated.
Sol) Figure 19 (currently Figure 15) has been modified to be simpler.
I suggest to the authors that the text on lines 168-170 be placed on line 166, so that table 1 is introduced and does not appear right after subheading 2.2.3.
Sol) Those parts have been modified as you recommend. Thank you.
Similarly, I suggest that figures be introduced first in the text and then included. For example, figure 4 I would introduce on line 191, figure 5 on line 223, figure 6 on line 236, figure 7 on line 271, figure 9 on line 322, figure 14 on line 453.
Sol) Those parts have been modified as you recommend. And another reviewer advised that sections 2.2.3 (abnormality of P, R, and T wave) and section 2.2.5 overlap. has been removed So section 2.2.3 has been removed. Thank you.
Thank you for your kindly comment and advice !!
Reviewer 2 Report
The article presents methods of detecting elements of the ECG signal.
Both the introduction, the description of the method and the research were properly described.
In the studies, data from the mit-bih were used.
I would like to point, that the signals from these databases have a quite good SNR.
It seems to me that it will be valuable to show the impact of noise with different characteristics (e.g. white, muscle) of different energy (e.g. for SNR = 0.5,10,15) on the obtained detection results.
These types of experiments will show the robustness of the proposed solutions to more diverse working conditions.
Author Response
Reviewer 2
The article presents methods of detecting elements of the ECG signal. Both the introduction, the description of the method and the research were properly described.
In the studies, data from the mit-bih were used. I would like to point, that the signals from these databases have a quite good SNR.
It seems to me that it will be valuable to show the impact of noise with different characteristics (e.g. white, muscle) of different energy (e.g. for SNR = 0.5,10,15) on the obtained detection results. These types of experiments will show the robustness of the proposed solutions to more diverse working conditions.
Sol) A noise experiment has been added as follows (in line 409~415 and Table 3).
The proposed R peak detection performance for noises is evaluated in Table 3. Tape 102 is a relatively clean signal with no noise, while date 105 contains large induced noises. For evaluation, the MATLAB function ‘awgn’, which adds white Gaussian noise to the signal, was used for a signal-to-noise ratio (SNR) of 0.5 to 80. The proposed method for noisy signals of 5 dB SNR or higher showed relatively good sensitivity and positive prediction performance, but the performance deteriorated rapidly for noise signals of 1 dB SNR or less.
Thank you for your kindly comment and advice !!
Reviewer 3 Report
In the paper "Electrocardiogram Fiducial Point Detector Using bilateral Filter and Symmetrical Point-filter Structure" the authors present a method to detect and delineate QRS complexes and P- and T-Waves applying 1D bilateral blurring filters and binary coding. They also demonstrate how their method facilitates and helps proper HRV analysis.
Recommendations:
======================
The paper is with 30 pages single column double spaced quite lengthy for a research paper which typically has about 10 to 12 two column single spaced pages. In the present form this would be exceeded by far. To shorten and straighten the paper i do suggest to drop the section 2.2.3 Abnormality of important waves.
It does not provide much information related to the proposed approach. Further its most important aspects are anyway repeated in Sections 2.2.5 Detection of Abnormal R-peak through to 2.2.8 FP detection of P- and T-Waves. Further is its main message discussed in Section 3 Results and section 5.
By the way section 4 Discussion is missing at least its section heading was accidentally deleted.
Figures in general:
Please select colours and line-style which also allow to follow your explanations when displaying, printing the paper in grey-scale mode.
Table 2:
I would drop as its information is quite low
Section 3.4 HRV analysis for MIT-DB:
Consider extracting it into a dedicated paper, possibly demonstrating that your method not only works for MIT-DB but also improves HRV analysis on datasets from other databases and real world data and shorten the presentation to only two to three prominent examples from different records as part of the discussion section.
Figure 19:
is to complex and each individual plot to tiny to be readable.
Section 2.2.4:
Use symbols for domain variance and range variance computed for the sliding windows which can clearly and unmistakeably distinguished from the variances globally selected for your filters.
Figure 7:
replace by formula.
Section 3.2:
Performance Evaluation of R-Peak Detection. Your criterion for when counting a detected R-Peak as TP when annotation from MIT-DB record is within +-100ms is quite coarse and inaccurate. In recent literature about +- 50 between Annotation and detected peak is used. As your approach is also delineating Q and Speaks and their onsets as well as split/fractionated R peaks the Method used by Thurner et Al 2020 "Complex-Pan-Tompkins-Wavelets: Cross-Channel ECG
Beat Detection and Delineation" would best suite your purpose. It is aware of the longer duration of split/fractionated - R peaks and can handle cases where the Annotation is placed on the maximum preceding or following the location your method selected. At the same time it is less fooled by noise and pacemaker peaks immediately preceding or following QRS Complexes or even partially shading parts of the PQRST sequence.
References/Citations:
Please reduce the amount of the literature cited and provide more recent results for example Thurner et Al (2020) Complex-Pan-Tompkins-Wavelets: Cross-Channel ECG
Beat Detection and Delineation", Böck et al (2015) “Global Decision Making for Wavelet Based ECG Segmentation”, Chen et al (2020) “A crucial wave detection and delineation method for twelve-lead ecg signals” or Soe et al (2020) “Ecg signal classification using discrete wavelet transform and pan tompkins algorithm” in the discussion and comparison of your proposed work.
All of them do delineate PQRST segments on single channel data or across multiple channels and are able to detect of split/fractionated R-Peaks . Currently the far majority of technical work cited by the authors is from 2010 and older only about 15 publications are from between 2011 and 2018. Especially the claim that the presented method is the only one so far able to detect and delineate split/fractionated R-Peaks properly is not valid, considering more recent publications on single and multi channel approaches. So I do consider a really thorough literature research as mandatory for successful publication.
English Language:
The presentation is quite well written. Never the less I do suggest to let an English native or somebody really firm in technical English proof it to ensure minor glitches and literal translation form other language reduce quality of paper.
Author Response
Reviewer 3
In the paper "Electrocardiogram Fiducial Point Detector Using bilateral Filter and Symmetrical Point-filter Structure" the authors present a method to detect and delineate QRS complexes and P- and T-Waves applying 1D bilateral blurring filters and binary coding. They also demonstrate how their method facilitates and helps proper HRV analysis.
Recommendations:
======================
The paper is with 30 pages single column double spaced quite lengthy for a research paper which typically has about 10 to 12 two column single spaced pages. In the present form this would be exceeded by far. To shorten and straighten the paper i do suggest to drop the section 2.2.3 Abnormality of important waves.
It does not provide much information related to the proposed approach. Further its most important aspects are anyway repeated in Sections 2.2.5 Detection of Abnormal R-peak through to 2.2.8 FP detection of P- and T-Waves. Further is its main message discussed in Section 3 Results and section 5.
Sol) Section 2.2.3 (and the reference 59~73) has been removed as you recommend. Table 1 and its description are included in section 2.2.5. And lines 47 to 63 (and the reference 32~38) in the introduction and their references were deleted due to the low relevance to the goal of this paper.
By the way section 4 Discussion is missing at least its section heading was accidentally deleted.
Sol) Sorry. The conclusion is section 4. It has been fixed.
Figures in general: Please select colours and line-style which also allow to follow your explanations when displaying, printing the paper in grey-scale mode.
Sol) As per your advice, the color of the text and lines in Figure 8 has been corrected to black.
Table 2: I would drop as its information is quite low
Sol) As per your comment, Table 2 and the explanation was deleted.
Section 3.4 HRV analysis for MIT-DB:
Consider extracting it into a dedicated paper, possibly demonstrating that your method not only works for MIT-DB but also improves HRV analysis on datasets from other databases and real world data and shorten the presentation to only two to three prominent examples from different records as part of the discussion section.
Sol) Figure 15 and Table 8 have been modified to be simpler. It was reduced to meaningful 6 of the original 9 files. MIT-DB was used for HRV analysis because it contained the patient's heart disease and condition according to the measurement time. It was difficult to find a well-documented database of patient conditions over time of measurement. In addition, actual ECGs are currently difficult to obtain due to IRB issues.
Figure 19: is to complex and each individual plot to tiny to be readable.
Sol) In Figure 15, for the readability of the figure, 202N(806), 214L(1200), and 233N(186), which are of low importance among the experimental files, were deleted, and the heart rate trend and RR interval trend, which overlap in Table 8, were deleted from the figure.
Section 2.2.4:
Use symbols for domain variance and range variance computed for the sliding windows which can clearly and unmistakeably distinguished from the variances globally selected for your filters.
Sol) The symbols for domain variance and range variance have been changed in Equation (3)~(5).
Figure 7: replace by formula.
Sol) In line 164, the figure has been replaced with equation (2).
Section 3.2:
Performance Evaluation of R-Peak Detection. Your criterion for when counting a detected R-Peak as TP when annotation from MIT-DB record is within +-100ms is quite coarse and inaccurate. In recent literature about +- 50 between Annotation and detected peak is used. As your approach is also delineating Q and Speaks and their onsets as well as split/fractionated R peaks the Method used by Thurner et Al 2020 "Complex-Pan-Tompkins-Wavelets: Cross-Channel ECG Beat Detection and Delineation" would best suite your purpose. It is aware of the longer duration of split/fractionated - R peaks and can handle cases where the Annotation is placed on the maximum preceding or following the location your method selected. At the same time it is less fooled by noise and pacemaker peaks immediately preceding or following QRS Complexes or even partially shading parts of the PQRST sequence.
Sol) The R peak detection performance was newly simulated according to the standard of the existing literature, about +- 50 between Annotation and detected peak. In the new simulation results, one additional false positive occurred in tapes 117, 205, and 232. The new results are reflected in Table 2.
References/Citations:
Please reduce the amount of the literature cited and provide more recent results for example Thurner et Al (2020) Complex-Pan-Tompkins-Wavelets: Cross-Channel ECG Beat Detection and Delineation", Böck et al (2015) “Global Decision Making for Wavelet Based ECG Segmentation”, Chen et al (2020) “A crucial wave detection and delineation method for twelve-lead ecg signals” or Soe et al (2020) “Ecg signal classification using discrete wavelet transform and pan tompkins algorithm” in the discussion and comparison of your proposed work.
All of them do delineate PQRST segments on single channel data or across multiple channels and are able to detect of split/fractionated R-Peaks. Currently the far majority of technical work cited by the authors is from 2010 and older only about 15 publications are from between 2011 and 2018. Especially the claim that the presented method is the only one so far able to detect and delineate split/fractionated R-Peaks properly is not valid, considering more recent publications on single and multi channel approaches. So I do consider a really thorough literature research as mandatory for successful publication.
Sol) Section 2.2.3 (and the reference 59~73) has been removed as you recommend. Table 1 and its description are included in section 2.2.5. And lines 47 to 63 (and the reference 32~38) in the introduction and their references were deleted due to the low relevance to the goal of this paper. And the latest paper you recommended below has been added in line 475 and 489, Table 6, and Reference 48.
[48] Chen, G.; Chen, M.; Zhang, J.; Zhang, L.; Pang. C. A crucial wave detection and delineation method for twelve-lead ECG signals. IEEE Access. 2020, 8, 10707-10717.
English Language:
The presentation is quite well written. Never the less I do suggest to let an English native or somebody really firm in technical English proof it to ensure minor glitches and literal translation form other language reduce quality of paper.
Sol) The entire paper has been revised once again by a native speaker.
Thank you for your kindly comment and advice !!
Round 2
Reviewer 2 Report
The article has been amended following the recommendations of other reviewers.
The results of the experiment that I have suggested are presented.
However, apart from the tabular summary of the results, it would also be nice to see their graphical representation.
Reviewer 3 Report
In the paper "Electrocardiogram Fiducial Point Detector Using bilateral Filter and Symmetrical Point-filter Structure" the authors present a method to detect and delineate QRS complexes and P- and T-Waves applying 1D bilateral blurring filters and binary coding. They also demonstrate how their method facilitates and helps proper HRV analysis.
The authors have considered all recommendation and improved and streamlined the paper. It now a lot more obvious which applications and environments the presented method is aiming at.
This manuscript is a resubmission of an earlier submission. The following is a list of the peer review reports and author responses from that submission.